# Assessing the cloud radiative bias at Macquarie Island in the ACCESS-AM2 model

Zhangcheng Pei[1,2,3], Sonya L. Fiddes[1,2], W. John R. French[4,1], Simon P. Alexander[4,1], Marc D. Mallet[1], Peter Kuma[5], and Adrian McDonald[6]

[1]Australian Antarctic Partnership Program, Institute for Marine and Antarctic Studies, University of Tasmania, Hobart, Australia
[2]Australian Research Council Centre of Excellence for Climate Extremes, University of Tasmania, Hobart, Australia
[3]College of Oceanic and Atmospheric Sciences, Ocean University of China, Qingdao, China
[4]Australian Antarctic Division, Hobart, Australia
[5]Department of Meteorology, Stockholm University, Stockholm, Sweden
[6]School of Physical and Chemical Sciences, University of Canterbury, Christchurch, Aotearoa/New Zealand

**Correspondence:** Zhangcheng Pei (zhangcheng.pei@utas.edu.au)

**Abstract.** As a long-standing problem in climate models, large positive shortwave radiation biases exist at the surface over the Southern Ocean, impacting the accurate simulation of sea surface temperature, atmospheric circulation, and precipitation. Underestimations of low-level cloud fraction and liquid water content are suggested to predominantly contribute to these radiation biases. Most model evaluations for radiation focus on summer and rely on satellite products, which have their own limitations. In this work, we use surface-based observations at Macquarie Island to provide the first long-term, seasonal evaluation of both downwelling surface shortwave and longwave radiation in the Australian Community Climate and Earth System Simulator Atmosphere-only Model Version 2 (ACCESS-AM2) over the Southern Ocean. The capacity of the Clouds and the Earth's Radiant Energy System (CERES) product to simulate radiation is also investigated. We utilise the novel lidar simulator, the Automatic Lidar and Ceilometer Framework (ALCF) and all-sky cloud camera observations of cloud fraction to investigate how radiation biases are influenced by cloud properties.

Overall, we find an overestimation of $+9.5 \pm 33.5\,\mathrm{W\,m^{-2}}$ for downwelling surface shortwave radiation fluxes and an underestimation of $-2.3 \pm 13.5\,\mathrm{W\,m^{-2}}$ for downwelling surface longwave radiation in ACCESS-AM2 in all-sky conditions, with more pronounced shortwave biases of $+25.0 \pm 48.0\,\mathrm{W\,m^{-2}}$ occurring in summer. CERES presents an overestimation of $+8.0 \pm 18.0\,\mathrm{W\,m^{-2}}$ for the shortwave and an underestimation of $-12.1 \pm 12.2\,\mathrm{W\,m^{-2}}$ for the longwave in all-sky conditions. For the cloud radiative effect (CRE) biases, there is an overestimation of $+4.8 \pm 28.0\,\mathrm{W\,m^{-2}}$ in ACCESS-AM2 and an underestimation of $-7.9 \pm 20.9\,\mathrm{W\,m^{-2}}$ in CERES. An overestimation of downwelling surface shortwave radiation is associated with underestimated cloud fraction and low-level cloud occurrence. We suggest that modelled cloud phase is also having an impact on the radiation biases. Our results show that the ACCESS-AM2 model and CERES product require further development to reduce these radiation biases, not just in shortwave and in all-sky conditions, but also in longwave and in clear-sky conditions.

# 1 Introduction

The Southern Ocean (SO) makes up a significant component of the Earth's climate system. As one of the cloudiest regions on Earth, the SO strongly influences the global energy balance and climate (Trenberth et al., 2009; Gettelman et al., 2020). A considerable deficit of reflected shortwave radiation at the top of the atmosphere (TOA) and an excess of absorbed shortwave radiation at the surface over the SO has been identified in both climate models and reanalysis (Trenberth and Fasullo, 2010; Bodas-Salcedo et al., 2014; Kay et al., 2016; Zhang et al., 2016; Fiddes et al., 2022; Cesana et al., 2022; Mallet et al., 2023). Cloud biases tend to limit the capacity of coupled models to accurately derive sea surface temperatures (SSTs) (Hyder et al., 2018), atmospheric circulation (Ceppi et al., 2012) and precipitation (Hwang and Frierson, 2013), and to correctly predict future climate changes (Trenberth and Fasullo, 2010; McCoy et al., 2015). As a key driver of global climate, it is important that we unravel what causes these radiative biases over the SO. Previous studies have suggested that the poor representation of clouds in climate models is the main contributor to the radiative biases (Bodas-Salcedo et al., 2012; Franklin et al., 2013; Mason et al., 2015), as clouds primarily control the TOA and surface energy budgets in the climate system (Bennartz et al., 2013; Luo et al., 2016).

Novel techniques including simulators for both satellite retrievals and in-situ observations, which are vital for model evaluation, have been developed in recent years. The Cloud Feedback Model Intercomparison Project (CFMIP) Observation Simulator Package (COSP) was created to allow quantitative examination of cloud properties, humidity, and precipitation processes in various numerical models (Bodas-Salcedo et al., 2011). Kuma et al. (2021) more recently have developed the Automatic Lidar and Ceilometer Framework (ALCF) to make automatic lidar and ceilometer (ALC) data comparable with climate models, including both global climate models (GCM) and numerical weather prediction (NWP) models. Large networks of lidars and ceilometers have been installed globally, for instance, Cloudnet (Illingworth et al., 2007), E-PROFILE (Illingworth et al., 2019), and ARM (Campbell et al., 2002). However, surface-based ceilometer observations of cloud frequency of occurrence and cloud boundaries over the SO remain sparse (Kuma et al., 2020). The ALCF can utilize the enormous database of surface-based ceilometer observations to evaluate the cloud occurrence and cloud characteristics in models and reanalysis. This is accomplished by extracting two-dimensional profiles (time x height) from the model data, using a modified COSP lidar simulator to perform radiative transfer calculations, calibrating and resampling the observed attenuated volume backscattering coefficient to a common resolution, and conducting similar cloud detection on both the simulated and observed attenuated volume backscattering coefficient (Kuma et al., 2021).

Aside from these new evaluation techniques, a number of statistical methods have been applied to understand the contribution of clouds to the model radiation biases. Williams and Webb (2009) used a cloud clustering approach to establish cloud regimes in models and compared them with satellite observations, showing a positive bias of shortwave cloud radiative effect in models. Field et al. (2011) utilized the cyclone compositing method to illustrate the underestimation of the TOA reflected shortwave radiation on the cold-air side of cyclones in models. These two techniques were combined by Bodas-Salcedo et al. (2014) to relate cloud regimes and radiative biases to different climatic conditions. It was observed that the cold-air side of the cyclone

composite is where the majority of model biases appear, and they mostly occur in the midlevel cloud regime (Bodas-Salcedo et al., 2014).

By incorporating the observational simulators and statistical analysis, climate models' outputs can be assessed against those observations. From previous research on the evaluation of cloud property simulations in models, it can be summarised that, over the SO region, the simulated low-level cloud fractions tend to be lower than both satellite observations (Trenberth and Fasullo, 2010; Bodas-Salcedo et al., 2012; Franklin et al., 2013) and surface-based observations (Protat et al., 2017; Klekociuk et al., 2020; Wang et al., 2020). However, discrepancies do exist between surface and satellite observations due to limitations of near-surface cloud retrievals of satellite.

In the context of widespread supercooled liquid clouds (SLCs), the underestimation of liquid water content in the clouds causes less reflective clouds and consequently less reflected shortwave radiation in the model at TOA (Hu et al., 2010; Bodas-Salcedo et al., 2016). Additionally, the poor representations of cloud feedbacks attributed to the reduction in low cloud coverage and water content lead to higher climate sensitivity in the Coupled Model Intercomparison Project phase 6 (CMIP6) compared to the previous version (Zelinka et al., 2020; Schuddeboom and McDonald, 2021; Kuma et al., 2023). Failure to accurately simulate physical properties of clouds in climate models emphasizes the necessity to use a variety of observational datasets to fully evaluate the models and correct biases through modifying the simulation of cloud fraction, cloud types, and cloud thermodynamic phases.

Surface-based observations and satellite products are two main types of datasets used to assess the model's performance. Numerous satellite-based evaluations have been previously conducted (Bodas-Salcedo et al., 2012, 2014, 2016; Luo et al., 2016), including for the Australian Community Climate and Earth System Simulator (ACCESS) model (Fiddes et al., 2018, 2022). Tansey et al. (2022) examined surface precipitation measurements during MICRE and compared them with data from Cloud-Sat, revealing several notable differences attributable to satellite instrument sensitivities and algorithm structure. This indicates the limitations of satellites in observing low-level clouds over the SO, which serves as a strong motivation for utilizing ground-based observations to calibrate satellite products. Nonetheless, ground-based observations in the SO and Antarctica remain limited due to the harsh atmospheric environment and lack of remote sites for measurements (Lawson and Gettelman, 2014), leading to less advanced model evaluation techniques than for the Northern Hemisphere. The parameterisations of models have not been comprehensively developed or tuned for the SO region, on account of the paucity of comparable field observations and suitable tools that can allow one-to-one comparison between models and observations (McFarquhar et al., 2021; Kuma et al., 2021). In recent years, several campaigns have been conducted to collect cloud properties over the SO (Protat et al., 2017; McFarquhar et al., 2021; Kremser et al., 2021). Using these observational data to test climate models with the latest simulators and statistical analysis, as well as calibrate satellite data, remains a critical task.

In this work, we evaluated the capability of ACCESS Atmosphere-only Model Version 2 (AM2) to simulate the downwelling surface radiation, cloud radiative effect and limited cloud properties. Performance of the Cloud and the Earth's Radiant Energy System (CERES) product in reproducing surface radiation and cloud radiative effect was also assessed. The campaign described by McFarquhar et al. (2021) and Tansey et al. (2022) at Macquarie Island was used as the observational dataset for comparison.

Furthermore, the ALCF product was used to explore the connection of cloud occurrence and radiative biases in the ACCESS-AM2 model compared with ceilometers for the first time.

The structure of this paper is organized as follows: Section 2 describes the data and methods used in the study; Section 3 evaluates the surface radiative bias in the ACCESS-AM2 model and CERES product; Section 4 presents the distribution of cloud fraction and explores the relationship between cloud fraction and radiative bias in the ACCESS-AM2 model; Section 5 examines the histograms of cloud occurrence using ALCF and investigates the link between cloud occurrence and radiative bias in the ACCESS-AM2 model; and Section 6 summarizes the results.

## 2 Data and methods

### 2.1 Overview of ground-based observations

The observational data used in this manuscript originated from the Macquarie Island Cloud and Radiation Experiment (MICRE), conducted by the United States Department of Energy (DOE) Atmospheric Radiation Measurement (ARM) program, the Bureau of Meteorology (BoM) and the Australian Antarctic Division (AAD), between March 2016 and March 2018. Located at 54.5°S, 158.9°E and with an altitude of 6 m (Figure 1a), the year-round AAD research station at Macquarie Island supports a range of scientific activities and has a long history of surface meteorology observations (Wang et al., 2015). The primary goal of MICRE was to gather surface-based measurements of radiation, precipitation, boundary layer (BL) clouds, and aerosol characteristics in order to evaluate satellite products and improve understanding of diurnal and seasonal fluctuations, particularly in terms of BL cloud vertical structure over the SO (McFarquhar et al., 2021). The data collected during the campaign includes downwelling surface radiation fluxes, precipitation rates, and ceilometer backscatter measurements along with standard meteorological observations.

### 2.2 Instrumentation

Instruments involved in the analysis of cloud radiative bias include a set of AAD broadbrand radiometers, which measure downwelling surface shortwave (SW) & longwave (LW) radiation fluxes; a ceilometer from University of Canterbury to determine cloud base height (CBH); and an AAD all-sky cloud camera to record cloud fraction. Measurements of all instruments cover the period from 5-April-2016 to 6-March-2018.

#### 2.2.1 Radiometers

Both a Kipp & Zonen CMP21 pyranometer (SW) and a Kipp & Zonen CGR4 pyrgeometer (LW) which are sensitive over 285-2800 nm and 4.5-42 $\mu$m respectively, were used to collect radiation data (Figure 1b). The sensors have a time resolution of 1 minute, and results were recorded as means and standard deviations for each of the 600 individual readings of output voltage at 1 minute interval, and logged on a Campbell Scientific CR3000 data logger.

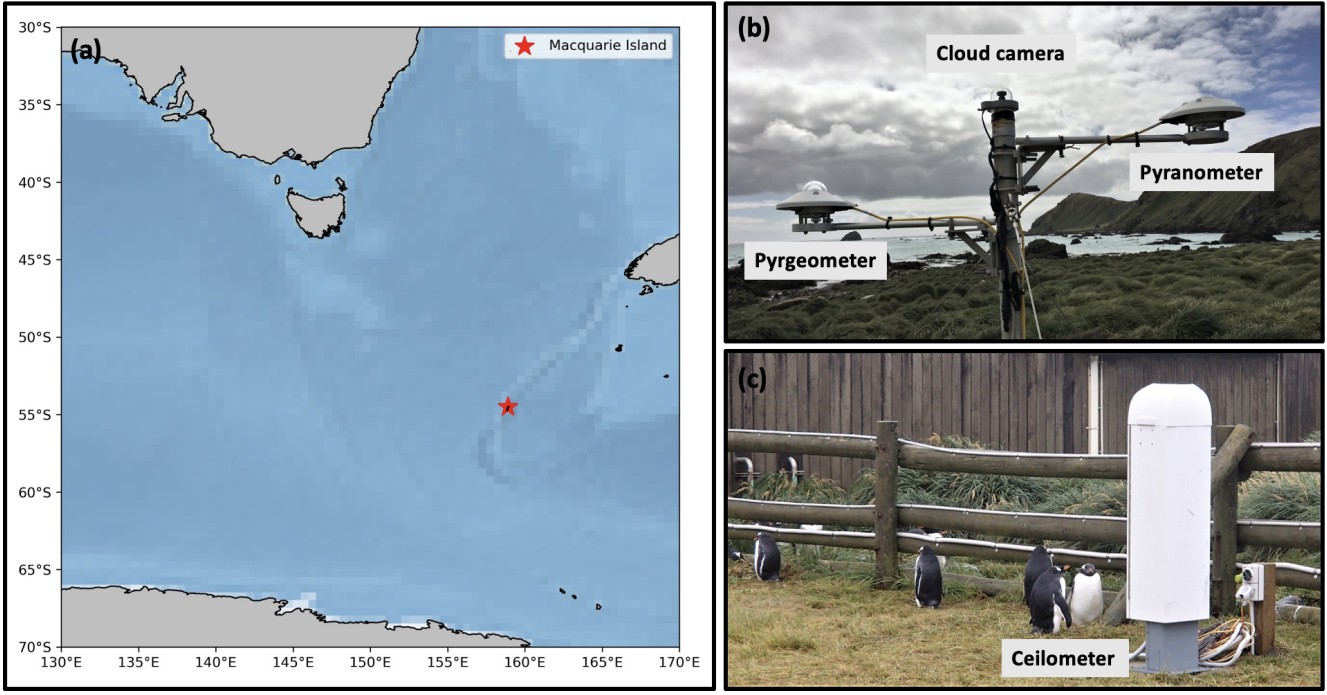

**Figure 1.** (a) Location of Macquarie Island (54.5°S, 158.9°E). The blue color scale represents the bathymetry of oceans. (b) Photo of cloud camera (on top of the mast), pyranometer (on the arm to the right), and pyrgeometer (on the arm to the left) installed in the Clean Air Lab enclosure (credit: Andrew Klekociuk, Australian Antarctic Division). (c) Photo of the Vaisala CL51 ceilometer installed in the Bureau of Meteorology lab, about 200m away from cloud camera and radiometers (credit: Jeff Aquilina, Bureau of Meteorology).

The LW radiation fluxes ($\mathrm{W\,m^{-2}}$) were calculated using:

$$L_d = \frac{U_{emf}}{S_L} + 5.67 \cdot 10^{-8} \cdot T_b^4 \tag{1}$$

where $U_{emf}$ is the pyrgeometer output voltage ($\mu V$), $S_L$ is the pyrgeometer sensitivity ($\mu V/(\mathrm{W\,m^{-2}})$), and $T_b$ is the thermistor temperature ($K$) of the pyrgeometer. The temperature ($K$) was calculated using:

$$T_b = (\alpha + [\beta \cdot ln(R) + \gamma \cdot (ln(R))^3])^{-1} \tag{2}$$

where R is the resistance ($\Omega$) and $\alpha$: $1.0295 \times 10^{-3}$, $\beta$: $2.391 \times 10^{-4}$, $\gamma$: $1.568 \times 10^{-7}$ are calibration coefficients from the Kipp & Zonen calibration certificate.

The SW radiation fluxes ($\mathrm{W\,m^{-2}}$) were calculated using:

$$S_d = \frac{U_{emf}}{S_S} \tag{3}$$

where $U_{emf}$ is the pyranometer output voltage ($\mu V$) and $S_S$ is the pyranometer sensitivity ($\mu V/(\mathrm{W\,m^{-2}})$).

The uncertainty in pyranometer measurements is derived from the sensitivity ($\pm$ 0.11 $\mu V/(\mathrm{W\,m^{-2}})$), and the uncertainty of the pyrgeometer is derived from the combination of sensitivity ($\pm$ 0.30 $\mu V/(\mathrm{W\,m^{-2}})$) and temperature measurements ($\pm$ 0.11 $K$). The radiometers were validated against a separate set of radiometers that were part of the ARM deployment at Macquarie Island (see Appendix A). The two independent data sets were found to be comparable within 2% for the SW and 5% for the LW, which are within the uncertainty of the instrumentation.

Both the pyranometer and pyrgeometer sensors were changed on 19-Mar-2017. Prior to this date the sensitivities ($S_S$ and $S_L$) were 8.89 and 13.01 $\mu V/(\mathrm{W\,m^{-2}})$ and after this date were 9.23 and 9.07 $\mu V/(\mathrm{W\,m^{-2}})$. From Equation 1, the pyrgeometer requires a temperature measurement to calculate the radiation flux. This is nominally obtained from a thermistor onboard the sensor, however a cable fault between 5-Jul-2016 14:37UT and 23-Nov-2016 02:22UT affected the thermistor resistance and consequently the measured temperature. Over this interval temperatures were substituted with those obtained from a similar thermistor onboard the pyranometer. Temperatures differences were within 1% between the two thermistors on average. Prior to 23-May-2017, the dataset was recorded with the Campbell logger default datatype FP2 which has a range limit of -7999 to 7999 $\mu V$. This inadvertently clipped the SW (pyranometer) data that exceeded 7999/8.89 = $\sim 900\,\mathrm{W\,m^{-2}}$ before 19-Mar-2017, and 7999/9.23 = $\sim 867\,\mathrm{W\,m^{-2}}$ between 19-Mar-2017 and 23-May-2017. The LW (pyrgeometer) data was unaffected by this effect. The limited clipped points accounted for approximately 3% of the whole dataset were removed. This was corrected to an IEEE 4-byte datatype on 23-May-2017, which has a $\pm 2.1^5 e^9$ range limit and 1 bit resolution which covered all levels voltage output by the sensor. Additionally, nine days of data, which accounted for approximately 1% of the whole dataset, were removed because of too few data points on those days to statically calculate a daily average.

### 2.2.2 Ceilometer

A Vaisala CL51 ceilometer, which is a vertically pointed near-infrared lidar with a regular vertical resolution of 10 m that operates at a wavelength of 910 nm ($\pm$ 10 nm) up to a range of 15.4 km, was employed to detect attenuated backscatter (Figure 1c). A two-dimensional (time × range) range-corrected attenuated backscatter profile was sampled every 6 seconds as the primary output (Klekociuk et al., 2020). The ceilometer observations were sub-sampled to 5-minute time resolution and 50-meter vertical resolution by averaging multiple columns and bins through ALCF (Kuma et al., 2020). Columns and bins here are respectively time and vertical intervals of the backscatter profile. Information on CBH, precipitation, and at times boundary layer height can be obtained from the backscatter profile using detection algorithms. Fog can be observed in the backscatter profiles as well. However, there are limitations to the capabilities of a ceilometer. Cloud tops and upper cloud layers are typically not visible in the backscatter profile due to the absorption of laser energy by thick clouds. As a result, the instrument is best suited for monitoring low-level clouds, although it may also be used to observe mid- to high-level clouds in the absence of low-level clouds (Klekociuk et al., 2020). Moreover, the signal and noise properties of the Vaisala CL51 ceilometer were investigated by Kotthaus et al. (2016) and a systematic bias was noted in the attenuated backscatter recorded by the instrument, which is determined by the internal calibration. Calibration of the instrument is achieved by scaling the backscatter signal to match the observed lidar ratio with the theoretical value (O'Connor et al., 2004).

During the selected period for conducting the radiation-cloud occurrence analysis in Section 5, which spanned from September 2017 to February 2018, approximately 6.7% of the ceilometer data were excluded due to poor quality.

### 2.2.3 All-sky cloud camera

For the cloud fraction (CF) analysis, colour images were taken at 1-minute intervals with an all-sky camera (Figure 1b). Both 'All-Sky' and 'Zenith' regions-of-interest (ROI) were included in the data processing, which comprised most of the unobstructed sky and an 8° radius field at the zenith, respectively. Based on a colour charge-coupled device (CCD) sensor, the camera contains a three-element 1.24 mm F2.8 lens that gives a 190° hemispherical "fisheye" field of view (FoV) to determine cloud distribution (Klekociuk et al., 2020; Wang et al., 2020). In terms of FoV of the cloud camera, it covered an area of 52 km in diameter at 4.5 km altitude. For each image captured during the day (solar elevation > 5°), a modified version of blue-red pixel ratio and differencing algorithms were employed to distinguish clear-sky and cloudy-sky pixels. Cumulative pixel counts, previously applied in several studies, were used to establish a CF (Li et al., 2011; Ghonima et al., 2012; Yabuki et al., 2014). For the pixel ratio algorithm (BdR - Blue channel divided by Red channel), a threshold of 1.3 was applied to the 8-bit (0–255) blue/red components to differentiate blue (clear-sky) pixels. For the pixel differencing algorithm (BmR - Blue channel minus Red channel), a threshold of 30 was applied to 8-bit (0-255) blue-red components to differentiate blue (clear-sky) pixels. The cloud camera dataset was organized to align with the available radiometer dataset, ensuring that the measurement of CF could be directly linked with radiation data.

## 2.3 Algorithms for cloud radiative effect and clear-sky radiation

### 2.3.1 Cloud radiative effect (CRE)

The cloud radiative effect (CRE) is defined as the influence of clouds on total radiation budget, computed from the difference in SW radiation and LW radiation between all-sky and clear-sky conditions (Wang et al., 2020). According to Shupe and Intrieri (2004) and Dommenget and Flöter (2011), the CRE is defined as:

$$CRE(\theta) = (1 - \alpha) \cdot (S(\theta) - S_0(\theta)) + \varepsilon \cdot (L(\theta) - L_0(\theta)) \tag{4}$$

which can be divided into shortwave cloud radiative effect (CRE$_{SW}$):

$$CRE_{SW}(\theta) = (1 - \alpha) \cdot (S(\theta) - S_0(\theta)) \tag{5}$$

and longwave cloud radiative effect (CRE$_{LW}$):

$$CRE_{LW}(\theta) = \varepsilon \cdot (L(\theta) - L_0(\theta)) \tag{6}$$

where $\alpha$ is the surface SW albedo, $\varepsilon$ is the LW surface emissivity, $\theta$ is the solar zenith angle, $S(\theta)$ and $S_0(\theta)$ are respectively the downwelling surface SW radiation in all-sky and clear-sky conditions, and $L(\theta))$ and $L_0(\theta)$ are respectively the down-

welling surface LW radiation in all-sky and clear-sky conditions. In this analysis, $\alpha = 0.055$ and $\varepsilon = 0.97$ were used to permit comparisons with earlier investigations (Fairall et al., 2008; Protat et al., 2017; Klekociuk et al., 2020).

### 2.3.2 Clear-sky radiation

Along with the measured SW and LW radiation under all-sky conditions, estimating the clear-sky radiation field is necessary to obtain the values of $S_0(\theta)$ and $L_0(\theta)$ before calculating the CRE using Equation 4. Macquarie Island is almost constantly covered by clouds, where only 0.6 % of time were classified as clear-sky by the all-sky camera. The limited observed clear-sky conditions meant we were unable to satisfactorily validate clear-sky models such as the SW clear-sky model by Corripio (2003) and the LW clear-sky model by Idso (1981). Both these clear-sky models, upon comparison to the ACCESS-AM2 and satellite

products, showed large biases, even using the parameters tuned for the SO provided by Wang et al. (2020).

With this in mind, we have used the downwelling surface clear-sky radiation fields from the European Center for Medium-range Weather Forecasting (ECMWF) Reanalysis 5 (ERA5) (Hersbach et al., 2020) for calculating cloud radiative effects. Assimilated measurements from different microwave sounders provide information on brightness temperatures and humidity to derive the clear-sky radiation in ERA5 (Hersbach et al., 2020). The ERA5 clear-sky fields have been used to validate

other clear-sky models, such as in Shakespeare and Roderick (2021). The ACCESS-AM2 and CERES products both take into account ERA5 atmospheric properties and hence each of these three products showed minimal differences. In this way we are able to limit introduced biases due to inaccurate clear-sky fields. We suggest that further efforts are needed to validate clear-sky models for the SO.

### 2.4 ACCESS-AM2

ACCESS-AM2 uses the same configuration as the ACCESS-CM2 (coupled model) without the ocean. The atmospheric component of ACCESS-AM2 is based on the UK Met Office's (UKMO) Unified Model (UM) version 10.6 Global Atmosphere (GA) 7.1 (Walters et al., 2019), with the Community Atmosphere Biosphere Land Exchange (CABLE) version 2.5 land surface model (Bi et al., 2020). The model has been operated globally at N96 resolution (approximately $1.25°$ latitude by $1.875°$ longitude) with 85 vertical levels (Bi et al., 2020; Bodman et al., 2020). Model output has been saved as daily means from

April 2016 to March 2018, and limited hourly instantaneous output from September 2017 to February 2018 to coincide with three other campaigns described in McFarquhar et al. (2021) besides MICRE.

The ACCESS-AM2 model is configured for the Atmospheric Model Intercomparison Project (AMIP) simulations, contributing to the Coupled Model Intercomparison Project phase 6 experiments (CMIP6) (Eyring et al., 2016). The model used in this study is nudged to ERA5 (Hersbach et al., 2020). The horizontal wind and temperature in the free troposphere and stratosphere

were nudged at every dynamical time step using reanalysis fields and updated every three hours (Fiddes et al., 2022). Sea surface temperatures (SSTs) and sea ice concentrations (SICs) are derived in accordance with the input4MIPS database and updated to cover the time period of this simulation (Hurrell et al., 2008). Solar forcing, greenhouse gases (GHGs), volcanic aerosol optical depth, and ozone are prescribed in the ACCESS-AM2 following the CMIP6 AMIP model configuration (Eyring et al., 2016).

Of interest to this study, the ACCESS-AM2 model uses the the Suite of Community RAdiative Transfer codes based on Edwards and Slingo (SOCRATES) (Edwards and Slingo, 1996) and Wilson et al. (2008)'s prognostic CF and condensate cloud scheme, which includes large-scale as well as convective clouds. For comparison with the observational data, radiation and prognostic CF in the model was linearly interpolated to the point nearest to Macquarie Island (54.5°S, 158.9°E). Additional details associated more generally with the ACCESS-AM2 model can be found in Bodman et al. (2020) and for these specific

simulations are detailed in Fiddes et al. (2022).

## 2.5   CERES SYN1° Dataset

The CERES project provides satellite-based observations of global clouds and radiation budgets. CERES instruments measure SW broadband radiances in 0.3-5 $\mu$m and LW broadband radiances in 5-200 $\mu$m (https://ceres.larc.nasa.gov/instruments). The CERES Synoptic TOA and downwelling surface fluxes and clouds (SYN) 1° product calculates hourly, 3-hourly, daily, and

monthly surface SW and LW fluxes using cloud and aerosol properties derived from a variety of sources (Rutan et al., 2015). In this study we examine the daily CERES SYN 1° Edition 4A product by linearly interpolating to the point nearest to Macquarie Island (54.5°S, 158.9°E) from April 2016 to March 2018, for consistency with the observational data.

## 2.6   ALCF

The ALCF is an open-source command line tool that processes ALC data and compare it to GCMs, NWP models and reanalysis.

It conducts the required steps to model the ALC attenuated volume backscattering coefficient by extracting cloud liquid and ice mixing ratios, cloud fraction, and thermodynamic data from the model. Additionally, the ALCF transforms the observed raw ALC attenuated volume backscattering coefficient profiles to make them comparable with the simulated profiles (Kuma et al., 2021).

    For the model data, ALCF first extracts two-dimensional cloud liquid and ice content profiles at the survey area, then uses

Subgrid Cloud Overlap Profile Sampler (SCOPS) to generate 10 random subcolumns for each profile to detect clouds in the model (Chepfer et al., 2008). The default setting for generating cloud overlap is maximum-random overlap assumption, which assumes neighboring layers with non-zero CF are fully overlapped, while layers separated by zero CF are randomly overlapped. The same sampling rate (5 min) and vertical bins (50 m) were used in lidar simulator to make the model and observations comparable. The attenuated volume backscattering coefficient profiles are then simulated for 10 subcolumns based on the

COSP lidar simulator. Subsequently, ALCF re-samples the observational profiles to increase the signal-to-noise ratio, subtracts the noise, calculates the lidar ratio, applies an absolute calibration, and uses a cloud detection algorithm to calculate cloud mask and CBH for both simulated and observational data. A threshold of $6 \times 10^{-6}$ $\mathrm{m}^{-1}\,\mathrm{sr}^{-1}$ for backscattering coefficient is applied to identify cloud mask after removing 5 standard deviations of range-scaled noise. This value was found to be a good compromise between false detection and misses of cloud at Macquarie Island, where the boundary layer aerosol is prevalent,

after testing different threshold values. This step is important to make the simulated and observed backscattering coefficient profiles comparable. Next, the statistical summary including CF, cloud frequency of occurrence (CFO) and attenuated volume backscattering coefficient histograms are derived. The CFO is calculated for each height level by counting the number of bins

which have a positive cloud mask divided by the total number of columns in the time range. The total CF is calculated by counting the number of columns which have at least one cloudy bin, divided by the total number of columns in the time range.

For the ceilometer data, ALCF applies the same operations as the model but starts from the denoised step. Plots of cloud occurrence representing the CBH and attenuated volume backscattering histogram are generated from the ALCF code. More information about this framework can be found in Kuma et al. (2021).

Several limitations exist within the ALCF that can cause uncertainties (Kuma et al., 2021). Firstly, the accuracy of the CL31 and CL51 ceilometers' calibration may be impacted by the absorption of water vapour at 910 nm, which can limit the precision of their comparison. However, it is improbable that the calculated cloud masks will be significantly influenced due to the high backscattering caused by clouds. Secondly, precipitation and aerosol are not currently implemented in the simulator. The cloud detection algorithm typically identifies observed precipitation as "cloud", whereas the simulated profile does not show any backscattering in the area where precipitation is occurring. Finally, the ALCs also encounter several measurement limitations. Specifically, inadequate overlap, dead time, and after-pulse corrections often yield sub-optimal outcomes at close range. Semi-automated methods include calculating the distribution of integrated attenuated volume backscattering coefficient by analyzing the height where maximum backscattering occurs.

The ALCF was operated from September 2017 to February 2018 to correspond with the hourly ACCESS-AM2 output in this study. The cloud occurrence in the ALCF output was primarily used to investigate the relationship between the cloud radiative bias and the representation of cloud occurrence in the model.

## 3 Surface radiative biases

### 3.1 All-sky and clear-sky surface radiation biases

Figure 2 shows the timeseries of daily-averaged surface SW and LW radiation fluxes at Macquarie Island from April 2016 to March 2018 based on the surface radiometer (black dotted line), ACCESS-AM2 model (red line), and CERES satellite product (blue line). During this two-year period, the SW radiation fluxes in the upper panel present a clear annual cycle, reaching the peak in austral summer (DJF) of around $250\,\mathrm{W\,m^{-2}}$. This annual cycle is also found in the magnitude of the SW fluctuations, with the smallest amplitude variability in winter and the largest amplitudes in summer. The surface SW radiation fluxes simulated by ACCESS-AM2 model and CERES align with observations regarding the $R^2$ values of 0.79 and 0.93 respectively (Figure 2a). For LW radiation fluxes in the lower panel, some variation is visible with lower downwelling LW flux in winter than in summer, which would be expected since the clouds and atmosphere are colder in winter and thus radiating less LW radiation to the surface. The magnitude of LW fluxes varies mainly between 250 to $350\,\mathrm{W\,m^{-2}}$, with a lower variability than SW fluxes. For the LW radiation fluxes, with the exception of winter (JJA) when the CERES exhibits a clear underestimation, the model and satellite conform to the observations well.

The model and satellite product respectively show Pearson correlations of SW radiation fluxes of 0.92 (ACCESS-AM2) and 0.98 (CERES) compared to the observations, with the periodicity of SW radiation enhancing these results. After monthly detrending, the correlations decrease to 0.72 and 0.94, suggesting a good performance by the model and excellent performance

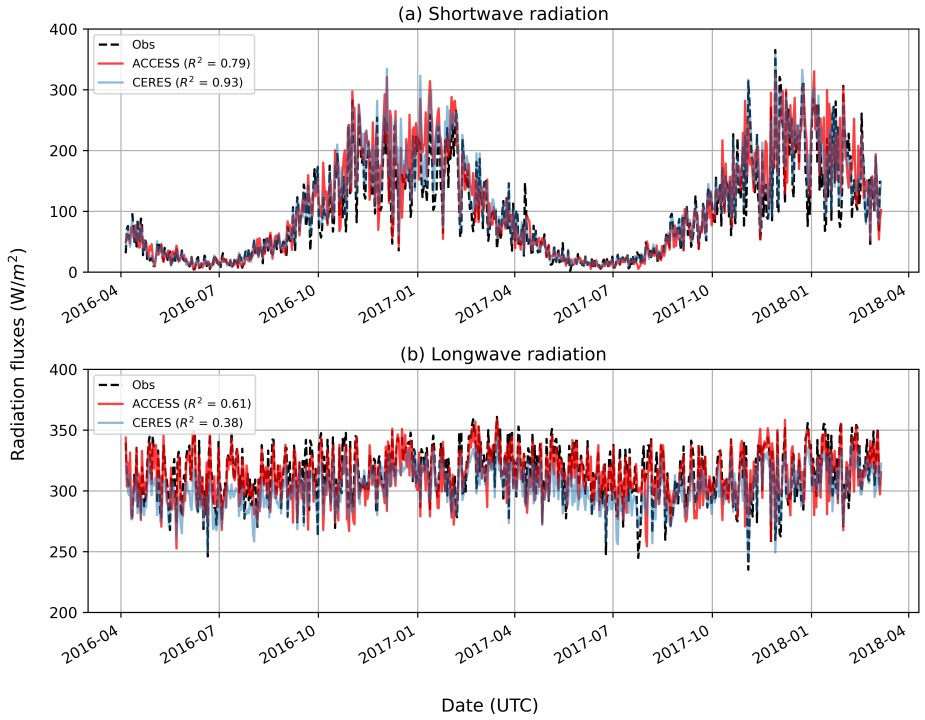

**Figure 2.** Timeseries of daily means of downwelling surface SW (a) and LW (b) radiation fluxes during MICRE. Black dotted line represents surface observations, red line represents ACCESS-AM2 outputs and blue line represents CERES observations. Coefficient of determination is indicated in the legend.

by the satellite product. The LW correlation between observation and ACCESS-AM2 model remains unchanged at 0.80 before and after eliminating monthly effects. However, this correlation rises from 0.82 to 0.86 between observation and satellite, possibly by artificially removing the winter-time low bias. The differing capability of model and satellite to simulate observed surface SW and LW demonstrates the necessity for validation of satellite products in reproducing surface radiation fluxes, including the radiative retrieval algorithms, before utilizing them to evaluate climate models.

Table 1 displays the total and seasonal averages of surface SW and LW radiation fluxes calculated using daily means under all-sky and modelled clear-sky conditions as well as their biases in observational, model, and satellite datasets. Figure 3 shows the seasonal distribution of SW and LW radiation fluxes in all-sky and clear-sky conditions. For the ACCESS-AM2 model, annually there is an overestimation of $+9.5 \pm 33.5\,\mathrm{W\,m^{-2}}$ in SW fluxes and a small underestimation of $-2.3 \pm 13.5\,\mathrm{W\,m^{-2}}$ in LW fluxes in cloudy conditions (Table 1). The number following the $\pm$ sign indicates the daily standard deviation but not the confidence interval as illustrated in the bolded brackets of the table. The overestimation of SW radiation in the model is

**Table 1.** Annual and Seasonal Means of Downwelling SW and LW Fluxes in all-sky and clear-sky conditions.

| $\mathrm{W\,m^{-2}}$ | Observation | ACCESS-AM2 | **Bias (ACCESS-AM2)** | CERES | **Bias (CERES)** |
|---|---|---|---|---|---|
| Annual | | | | | |
| SW mean | 95.9 [76.3] | 105.4 [83.4] | **9.5 [4.3]**\* | 103.9 [81.4] | **8.0 [4.2]**\* |
| SWcs mean | 189.5 [127.9] (ERA5) | 187.2 [127.3] | **-2.3 [6.8]** | 189.2 [127.6] | **-0.3 [6.8]** |
| LW mean | 314.0 [21.8] | 311.7 [19.8] | **-2.3 [1.1]**\* | 301.9 [16.6] | **-12.1 [1.0]**\*\*\* |
| LWcs mean | 251.0 [14.8] (ERA5) | 255.5 [15.6] | **4.5 [0.8]**\*\*\* | 255.3 [13.0] | **4.3 [0.7]**\*\*\* |
| Summer (DJF) | | | | | |
| SW mean | 171.5 [64.3] | 196.5 [60.5] | **25.0 [6.6]**\*\*\* | 191.3 [60.9] | **19.8 [6.6]**\*\* |
| SWcs mean | 341.5 [43.5] (ERA5) | 339.2 [42.6] | **-2.3 [4.6]** | 340.9 [43.5] | **-0.6 [4.6]** |
| LW mean | 320.7 [21.0] | 318.0 [19.7] | **-2.7 [2.2]** | 312.7 [14.6] | **-8.0 [1.9]**\*\*\* |
| LWcs mean | 259.7 [12.2] (ERA5) | 264.8 [13.1] | **5.1 [1.3]**\*\*\* | 262.5 [10.7] | **2.8 [1.2]**\* |
| Autumn (MAM) | | | | | |
| SW mean | 51.9 [37.3] | 52.3 [33.0] | **0.4 [4.1]** | 52.9 [35.9] | **1.0 [4.2]** |
| SWcs mean | 105.7 [59.6] (ERA5) | 103.8 [59.6] | **-1.9 [6.9]** | 105.3 [60.0] | **-0.4 [6.9]** |
| LW mean | 317.6 [20.5] | 314.0 [19.4] | **-3.6 [2.3]** | 302.9 [14.9] | **-14.7 [2.1]**\*\*\* |
| LWcs mean | 254.3 [14.2] (ERA5) | 257.6 [14.4] | **3.3 [1.7]**\* | 258.8 [12.4] | **4.5 [1.5]**\*\* |
| Winter (JJA) | | | | | |
| SW mean | 26.8 [17.6] | 24.6 [16.3] | **-2.2 [1.8]** | 27.1 [16.9] | **0.3 [1.8]** |
| SWcs mean | 52.5 [25.5] (ERA5) | 50.7 [25.1] | **-1.8 [2.6]** | 52.1 [25.5] | **-0.4 [2.7]** |
| LW mean | 307.3 [21.5] | 307.2 [17.8] | **-0.1 [2.1]** | 290.5 [12.7] | **-16.8 [1.8]**\*\*\* |
| LWcs mean | 242.7 [13.4] (ERA5) | 246.9 [14.3] | **4.2 [1.4]**\*\* | 248.6 [12.1] | **5.9 [1.3]**\*\*\* |
| Spring (SON) | | | | | |
| SW mean | 127.8 [60.8] | 141.1 [59.3] | **13.3 [6.3]**\* | 137.5 [60.0] | **9.7 [6.3]** |
| SWcs mean | 249.6 [77.2] (ERA5) | 246.2 [76.7] | **-3.4 [8.1]** | 249.3 [76.8] | **-0.3 [8.1]** |
| LW mean | 311.2 [21.6] | 308.4 [20.4] | **-2.8 [2.2]** | 302.1 [15.8] | **-9.1 [2.0]**\*\*\* |
| LWcs mean | 248.0 [13.2] (ERA5) | 253.1 [14.9] | **5.1 [1.5]**\*\*\* | 252.0 [11.7] | **4.0 [1.3]**\*\* |

*Note*. All values have units of $\mathrm{W\,m^{-2}}$. The bolded biases were calculated based on mean surface fluxes (e.g. ACCESS-AM2 - Observation, CERES - Observation). When present, brackets "[]" show day-to-day standard deviation, while bolded brackets show standard error of mean difference, which reflects if the biases can be considered as significant at a certain confidence interval. The biases with '*' mean the p-value < 0.1, with '**' mean the p-value < 0.01, and with '***' mean the p-value < 0.001.

pronounced in spring and becomes more so in summer, during which season the mean SW radiation simulated by the model is $+25.0 \pm 48.0\,\mathrm{W\,m^{-2}}$ higher than the observations. As illustrated in Figure 3a, the model's distribution (blue) exhibits a large shift to higher radiation fluxes relative to the observation (red) in the summer. The differences for LW radiation fluxes in the model are minor throughout all seasons, reaching $-4\,\mathrm{W\,m^{-2}}$ in autumn, with smaller differences in all other seasons. The

300

CERES product has an overestimation of +8.0 ± 18.0 W m$^{-2}$ in SW radiation fluxes and a large underestimation of -12.1 ± 12.2 W m$^{-2}$ in LW radiation fluxes in all-sky conditions. Similar to ACCESS-AM2, the SW radiation biases of the satellite product are greater in the spring and summer than in the autumn and winter. From Figure 3a, the satellite's distribution (green) shows a large shift to higher value in comparison to the observation (red) in the summer, which is comparable to the model. The LW radiation biases of the satellite are much larger than those of the model, with the highest biases occurring in autumn and winter. This is especially evident in Figure 3b, where there is a very significant shift to smaller radiation fluxes in the distribution of the satellite data compared to the observation and model.

When it comes to simulated clear-sky conditions, the ACCESS-AM2 surface shortwave (SWcs) and longwave (LWcs) radiation fluxes were found to have biases of -2.3 ± 3.7 W m$^{-2}$ and +4.5 ± 5.3 W m$^{-2}$ in total (Table 1), when compared to the ERA5 clear-sky product. Non-significant negative SWcs biases in the model are consistent across all seasons and the distribution of SWcs of ACCESS-AM2 and ERA5 fit well (Figure 3c). The biases for LWcs fluxes in the ACCESS-AM2 model are statistically significant and more notable in spring and summer (Figure 3d). The satellite's LW biases and seasonal distributions in clear-sky conditions are similar to ACCESS-AM2 when comparing with ERA5, while the SW biases are more negligible ( Figure 3c, d). The significant differences of LWcs in model and satellite compared to ERA5 highlight the need for more validation and development of especially the LWcs models. The SWcs models show smaller and insignificant biases, indicating less uncertainty.

After quantifying the average biases and the seasonal distributions of radiation data from the ACCESS-AM2 model and the satellite product, we can now further explore the causes of these biases. Numerous studies have corroborated the overestimation of surface SW radiation in climate models, reaching a maximum in summer (Trenberth and Fasullo, 2010; Franklin et al., 2013; Mason et al., 2015; Luo et al., 2016). The larger quantity of solar radiation in the spring and summer compared to the autumn and winter causes the cloud bias in these periods to have a larger impact on the radiative balance (Luo et al., 2016; Fiddes et al., 2022). Underestimated CF and liquid water content in the model are believed to be the major explanation for this overestimation (Mason et al., 2015; Luo et al., 2016; Kuma et al., 2020). "Too few and too bright" low-level clouds were identified as the cause of this SW bias in CMIP5 models (Nam et al., 2012; Wall et al., 2017). Nevertheless, more recently, Schuddeboom and McDonald (2021) discovered the exact contrasting result in the CMIP6 simulations, which demonstrates the importance of prioritizing the low-level cloud simulation to enhance the SW radiative balance over the SO. The LW radiation biases can be expected to also originate largely from cloud occurrence and cloud microphysics biases, and to a lesser extent atmospheric temperature and humidity biases (Wild et al., 2001). The physical reason is a high emissivity of clouds compared to the atmosphere, so the surface is more radiatively coupled to clouds as opposed to the thermally very cold space. For downwelling surface radiation in clear-sky conditions, Wild et al. (2006) suggests earlier GCMs overestimated SWcs radiation due to a lack of suitable aerosol forcing and an overestimated water vapor absorption. The influence of aerosol in the representation of SWcs radiation in climate models has been confirmed by Ruiz-Arias et al. (2013). Nevertheless, ACCESS has been shown (unpublished) to underestimate radiation relevant aerosol over the SO, which does not fit with the underestimated SWcs bias here. Significantly, further work needs to be done to understand both aerosol as well as water vapour, which has been understudied over the SO to solve the SWcs biases in the ACCESS-AM2 model. The consistent biases in LWcs could

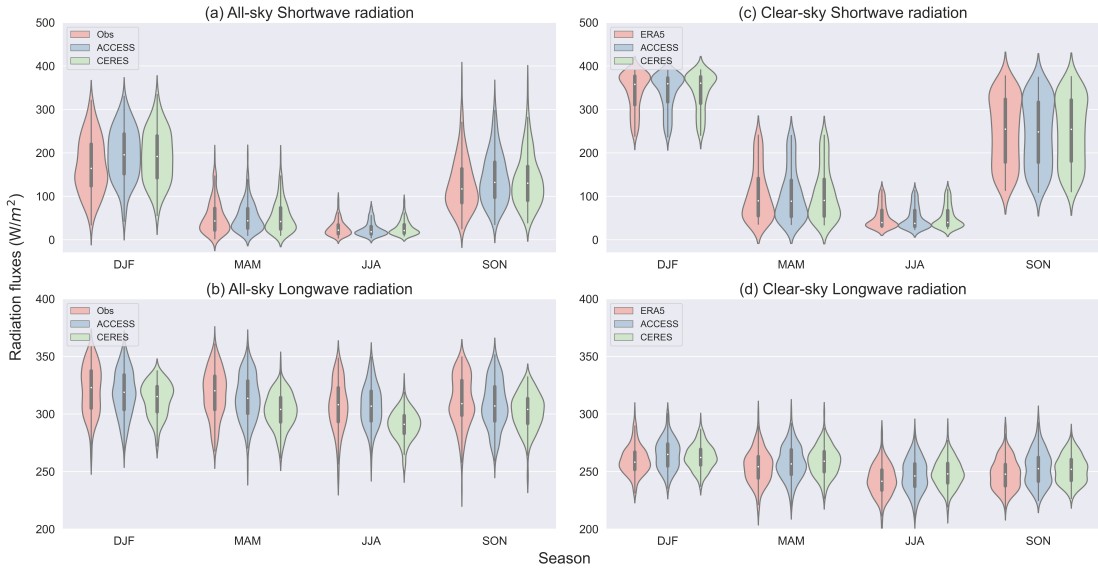

**Figure 3.** Violin plot of seasonal distributions of downwelling surface SW (a, c) and LW (b, d) radiation fluxes in all-sky and clear-sky conditions among surface observations, ACCESS-AM2 model, and satellite data. The white dot on the middle represents the median, the thick gray bar represents the interquartile range, and the thin gray line represents the rest of the distribution. The width of the violin plot represents the distribution of radiation value.

be partially explained by the difference in the humidity profiles and near-surface temperature according to Allan (2000), who evaluated the simulated LWcs against ground-based observations.

For CERES data, non-negligible biases are also present. The SW bias of $+8.0 \pm 18.0\,\mathrm{W\,m^{-2}}$ and LW bias of $-12.1 \pm 12.2\,\mathrm{W\,m^{-2}}$ are slightly different to the previous study by Hinkelman and Marchand (2020) in magnitude ($+10\,\mathrm{W\,m^{-2}}$ for SW bias and $-10\,\mathrm{W\,m^{-2}}$ for LW bias), which used a co-located observational dataset collected by ARM at Macquarie Island. These differences in SW & LW biases are possibly attributed to different temporal resolution of the CERES SYN product (hourly output used in Hinkelman and Marchand (2020) and daily output used in this study) and different interpolation methods to collocate data to Macquarie Island (Hinkelman and Marchand (2020) chose the nearest grid that contains Macquarie Island while this study linearly interpolated data to Macquarie Island). Other factors such as data gaps, sampling uncertainty, calibration offsets, different pyranometers, and local shadowing effects may also contribute to the biases difference. Hinkelman and Marchand (2020) showed that the LW biases in the CERES were caused by an inaccurately low CBH at night. Nonnegligible biases in SW and LW fluxes indicate the importance of evaluating and improving the retrieval algorithms for surface radiation fluxes in satellite data, with larger biases in summer for SW radiation and in winter for LW radiation.

Excellent alignment of SWcs radiation between the satellite and reanalysis is expected given the CERES product uses ERA5 to inform its radiative transfer algorithm. While few studies have been focused on LWcs biases, the biases of similar magnitude for the satellite product and the model suggest that more attention needs to be paid to the clear-sky algorithms, including for the ERA5 parameterisation. Once again, we suggest the parameterisation of humidity and temperature, and their use in the clear-sky models, must be a point of focus.

Understanding the biases in the respective SW and LW clear-sky biases is an important but often neglected component of understanding the CREs. Here we have shown that while the SWcs biases from ACCESS-AM2 and CERES (using similar meteorology driven by ERA5 and using the same method of calculating the clear-sky fluxes) are very similar, the same cannot be said for the LWcs. These differences, and how they affect the CRE, require further study. Wang et al. (2020) evaluated the cloud radiative effect of ERA5 using ship-based measurements in the SO during three summer seasons. Higher shortwave cloud radiative effect ($+77\,\mathrm{W\,m^{-2}}$) and lower longwave cloud radiative effect ($-18\,\mathrm{W\,m^{-2}}$) were detected in ERA5 in all-sky conditions, which are likely attributed to the higher occurrence of clouds over the Southern Ocean compared to what was modelled, and potentially resulting from the higher transmittance of clouds in the ERA5 (Wang et al., 2020). Regarding clear-sky conditions, no notable error was found in the ERA5 LW irradiance, while for SW, the observed values were $33\,\mathrm{W\,m^{-2}}$ higher than those predicted by ERA5. More recently, Mallet et al. (2023) found large downwelling SW radiation biases ($+54\,\mathrm{W\,m^{-2}}$) in the ERA5 compared with 25 years summertime surface measurements collected from ship and ground station over the SO. By employing machine learning techniques, cloud cover and relative humidity exhibited a strong contribution to the SW radiation biases. Despite these few studies on ERA5 radiation biases, a limited amount of research has been dedicated to investigating this issue, particularly in relation to clear-sky conditions. We suggest the importance of using ground-based observations of clear-sky radiation to evaluate the model and satellite, as well as validating the reanalysis product.

After investigating the SW and LW radiation biases of ACCESS-AM2 and CERES in both all-sky and clear-sky conditions, we next assess their capability to reproduce cloud radiative effects.

## 3.2 Surface cloud radiative effect (CRE) biases

The surface cloud radiative effect (CRE) determines the role of clouds in the surface radiation budget (defined in Section 2.3). A positive CRE indicates that clouds are warming the surface while a negative CRE implies a surface cooling.

Figure 4 shows the timeseries of daily average total CRE, $\mathrm{CRE}_{SW}$ and $\mathrm{CRE}_{LW}$ in the three datasets. The timeseries of total CRE and $\mathrm{CRE}_{SW}$ shows an evident annual cycle in which the value is significantly negative in the summer, reduces during autumn, and reaches a minimum value in winter. The $\mathrm{CRE}_{LW}$, similar to the all-sky condition, does not show a clear annual cycle, being stable across the year and ranging from 0 to $+80\,\mathrm{W\,m^{-2}}$.

Table 2 shows the value of the total and seasonal averages and biases of surface total CRE, $\mathrm{CRE}_{SW}$, and $\mathrm{CRE}_{LW}$ for surface-based observations, ACCESS-AM2 model and CERES product. Figure 5 shows the respective seasonal distributions. For the ACCESS-AM2 model, there is a total CRE of $-23.0 \pm 58.1\,\mathrm{W\,m^{-2}}$ contributed by a SW cooling of $-77.6 \pm 58.8\,\mathrm{W\,m^{-2}}$ and a LW warming of $+54.6 \pm 11.7\,\mathrm{W\,m^{-2}}$, during the 2-year period. A $\mathrm{CRE}_{SW}$ bias ($+11.2 \pm 31.1\,\mathrm{W\,m^{-2}}$) dominates the total CRE bias ($+4.8 \pm 28.0\,\mathrm{W\,m^{-2}}$). Figure 5a demonstrates that the total CRE bias is largest during the spring and summer.

**Table 2.** Annual and Seasonal Means of CRE, $CRE_{SW}$, and $CRE_{LW}$.

| $W\,m^{-2}$ | Observation | ACCESS-AM2 | Bias (ACCESS-AM2) | CERES | Bias (CERES) |
|---|---|---|---|---|---|
| **Annual** | | | | | |
| CRE mean | -27.8 [69.6] | -23.0 [58.1] | **+4.8 [3.4]** | -35.7 [56.3] | **-7.9 [3.4]**[*] |
| $CRE_{SW}$ mean | -88.8 [70.9] | -77.6 [58.8] | **+11.2 [3.5]**[**] | -80.9 [61.6] | **+7.9 [3.6]**[*] |
| $CRE_{LW}$ mean | +61.0 [12.4] | +54.6 [11.7] | **-6.4 [0.6]**[***] | +45.2 [11.5] | **-15.8 [0.6]**[***] |
| **Summer (DJF)** | | | | | |
| CRE mean | -101.6 [51.0] | -82.9 [43.3] | **+18.7 [5.0]**[***] | -92.4 [46.6] | **+9.2 [5.2]**[*] |
| $CRE_{SW}$ mean | -160.2 [58.7] | -134.4 [50.4] | **+25.8 [5.8]**[***] | -140.9 [52.1] | **+19.3 [5.9]**[**] |
| $CRE_{LW}$ mean | +58.6 [12.9] | +51.5 [11.4] | **-7.1 [1.3]**[***] | +48.5 [11.1] | **-10.1 [1.3]**[***] |
| **Autumn (MAM)** | | | | | |
| CRE mean | +10.7 [34.5] | +6.9 [32.4] | **-3.8 [3.9]** | -5.9 [28.9] | **-16.6 [3.7]**[***] |
| $CRE_{SW}$ mean | -50.8 [34.4] | -48.2 [33.0] | **+2.6 [3.9]** | -48.8 [30.5] | **+2.0 [3.8]** |
| $CRE_{LW}$ mean | +61.5 [11.0] | +55.1 [10.3] | **-6.4 [1.2]**[***] | +42.9 [10.2] | **-18.6 [1.2]**[***] |
| **Winter (JJA)** | | | | | |
| CRE mean | +38.4 [14.8] | +33.7 [14.6] | **-4.7 [1.5]**[**] | +16.5 [12.0] | **-21.9 [1.4]**[***] |
| $CRE_{SW}$ mean | -24.2 [13.5] | -24.7 [13.6] | **-0.5 [1.4]** | -24.1 [12.5] | **0.1 [1.4]** |
| $CRE_{LW}$ mean | +62.6 [11.6] | +58.4 [11.5] | **-4.2 [1.2]**[***] | +40.6 [9.7] | **-22.0 [1.1]**[***] |
| **Spring (SON)** | | | | | |
| CRE mean | -53.9 [55.4] | -45.7 [44.3] | **+8.2 [5.3]** | -57.1 [45.4] | **-3.2 [5.3]** |
| $CRE_{SW}$ mean | -115.1 [60.0] | -99.3 [49.8] | **+15.8 [5.8]**[**] | -105.7 [51.8] | **+9.4 [5.9]** |
| $CRE_{LW}$ mean | +61.2 [13.3] | +53.6 [12.4] | **-7.6 [1.3]**[***] | +48.6 [12.5] | **-12.6 [1.4]**[***] |

*Note*. All values have units of $W\,m^{-2}$. The bold values indicate the biases, which were calculated based on mean surface fluxes (e.g. ACCESS-AM2 - Observation, CERES - Observation). When present, brackets "[]" show day-to-day standard deviation, while bolded brackets show standard error of mean difference. The biases with '*' mean the p-value < 0.1, with '**' mean the p-value < 0.01, and with '***' mean the p-value < 0.001.

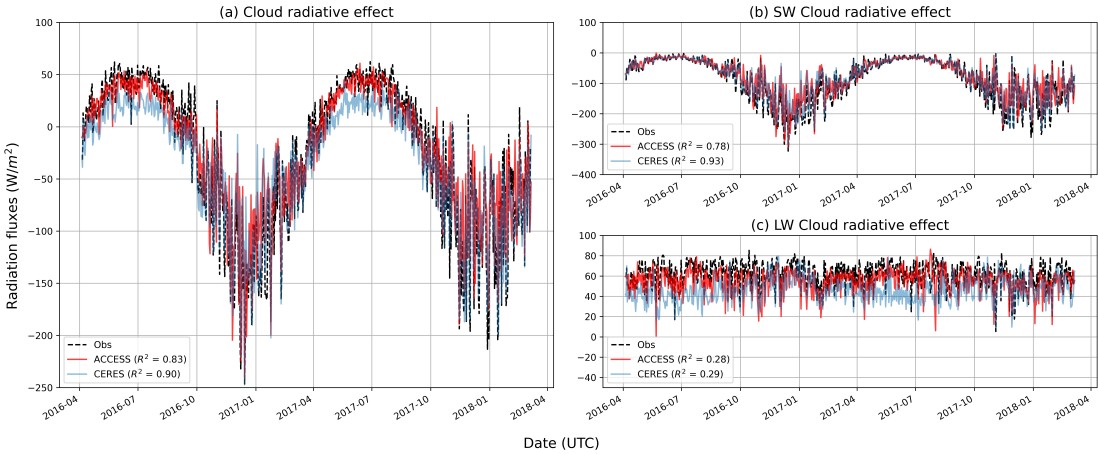

**Figure 4.** Timeseries of daily means of CRE (a), $CRE_{SW}$ (b) and $CRE_{LW}$ (c). Black dotted line represents surface observations, red line represents ACCESS-AM2 outputs and blue line represents CERES observations. Coefficient of determination is indicated in the legend.

During winter, when total CRE is at the most positive value, the $CRE_{LW}$ has a greater influence on the total CRE than the $CRE_{SW}$, mainly caused by the biases in clear-sky conditions (Table 1, 2, Figure 5). In comparison to ACCESS-AM2, CERES has a greater total CRE of -35.7 $\pm$ 56.3 W m$^{-2}$, which is attributed to a larger SW cooling of -80.9 $\pm$ 61.6 W m$^{-2}$ and a smaller LW warming of + 45.2 $\pm$ 11.5 W m$^{-2}$. The seasonal distribution of $CRE_{SW}$ biases in the CERES product follows the same pattern as the ACCESS-AM2 model (Table 2, Figure 5b). However, larger negative biases of $CRE_{LW}$ in autumn and winter than the model can be attributed to the large LW bias during these seasons in the satellite measurements (Table 1, Figure 5c). The overestimation of surface total CRE in ACCESS-AM2 is consistent with a series of studies (Allan, 2000; Protat et al., 2017; McFarquhar et al., 2021), with the source of overestimated surface SW radiation attributed to poor simulation of low-level cloud fraction and cloud liquid water content.

The diurnal cycle of CRE biases in summer was investigated in Appendix B. For ACCESS-AM2, the prevalent $CRE_{SW}$ biases contribute to the majority of the CRE bias, which peaks at noon and presents no bias at night. The $CRE_{LW}$ biases have lower magnitude than $CRE_{SW}$ and show no obvious variability (Figure B1). For CERES, $CRE_{SW}$ biases are comparable to ACCESS-AM2, while the negative $CRE_{LW}$ biases are substantial at local night (Figure B1), which has been attributed to wrong cloud base height (Hinkelman and Marchand, 2020). The diurnal cycle highlights that in summer the ACCESS-AM2 model is able to more accurately capture the characteristics of LW radiation than it can for SW radiation, and most of the cloud radiative biases can be attributed to poor simulation of SW radiation. While in CERES, poor SW simulation during the day and LW simulation during the night both contribute to the total cloud radiative biases.

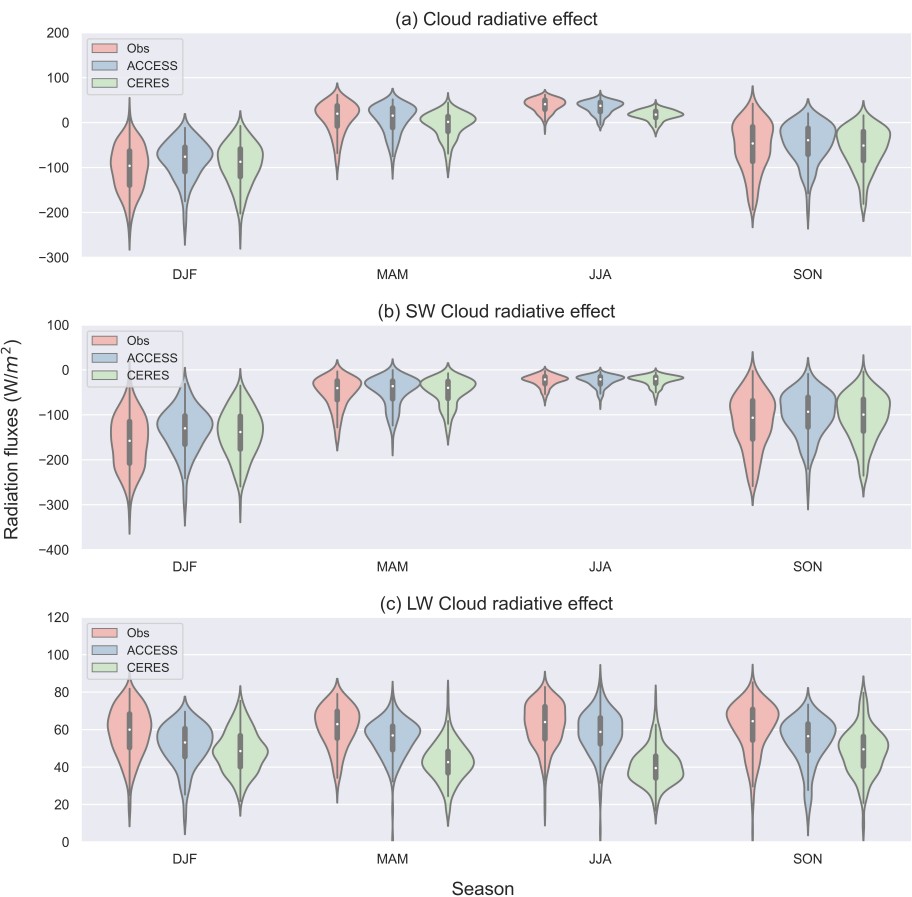

**Figure 5.** Same as Figure 3 but for CREs among surface observations, ACCESS-AM2 model, and satellite observations.

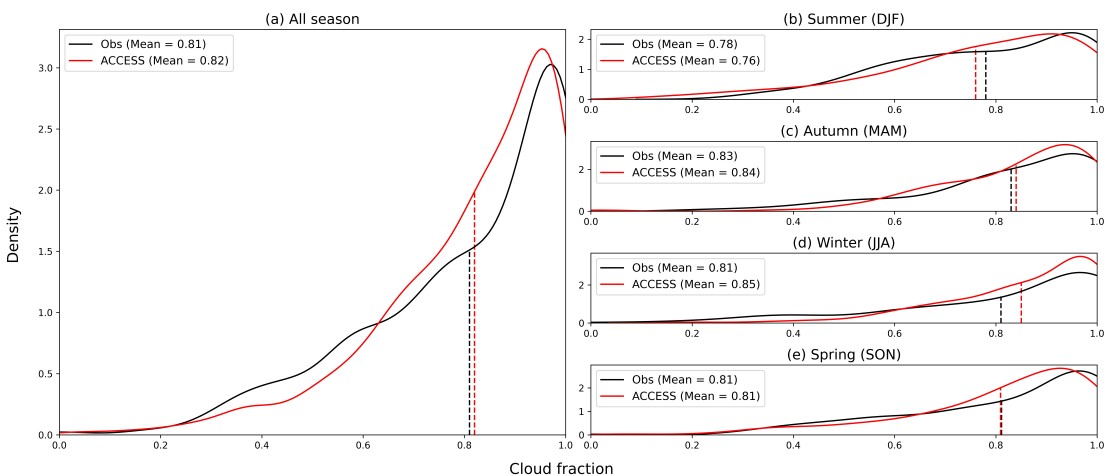

**Figure 6.** The total CF distribution (a) and seasonal CF distribution (b, c, d, e) of ACCESS-AM2 model and surface observations from April 2016 to March 2018. The vertical dashed lines indicate the mean of the datasets corresponding to their colour. Averages are shown in the bracket after the legend.

## 4 CF biases and their connection to radiative biases

After finding non-negligible radiative biases in the surface radiation and CRE in the ACCESS-AM2 model and CERES product, we now examine the cloud conditions associated with these biases. In line with theoretical expectation and previous studies that suggest CF has a significant influence on the downwelling surface radiation (Luo et al., 2016; Protat et al., 2017), we evaluate the distribution of CF and its relationship with radiative bias using daily observational data collected by cloud camera during MICRE from April 2016 to March 2018.

### 4.1 CF Distribution

The annual (a) and seasonal (b-e) CF distributions are shown in Figure 6. In the left panel, the daily averaged CF for the observations and model are respectively $0.81 \pm 0.19$ and $0.82 \pm 0.17$. The number following the $\pm$ sign here indicates the daily standard deviation of CF. The mean bias (ACCESS-AM2 – observation) is the integrated effect of an underestimated frequency of CF between 0.2 to 0.6 and an overestimated frequency of CF between 0.6 to 0.9 (Figure 6a). As shown in Figure 6e, the model accurately simulates the mean CF in the spring, but it still overestimates the CF frequency between 0.6 and 0.9. In both autumn and winter, the model overestimated the mean CF, with a more pronounced overestimation occurring in winter (Figure 6c,d). Unlike spring, autumn and winter, the summer mean CF in the model is lower than observed (Figure 6b).

Several previous studies examined the CF simulated by climate models or reanalysis in the SO and Antarctic regions during austral summer (Mason et al., 2015; Protat et al., 2017; Wang et al., 2020; McFarquhar et al., 2021). In the ACCESS1.3 model

for the high latitude SO (50°–65°S), Mason et al. (2015) found an overall CF deficit. Protat et al. (2017) discovered that the regional NWP version of the ACCESS model overestimates the frequency of intermediate CF, but underestimates the frequency of extremely low or extremely high CF over the SO. Wang et al. (2020) found an underestimation of daily averaged CF in the ERA5 datasets. In McFarquhar et al. (2021), the radiation bias in the NWP version of ACCESS was shown to link mostly to
low-level clouds.

During MICRE, the sky was overcast over almost the entire observation period (bar four days). As mentioned in the previous section, our study also finds a total overestimation for average downwelling surface SW radiation in the ACCESS-AM2 model. However, here we show mean overestimations in CF by the model in autumn and winter, a result of both an overestimation in the frequency of CF between 0.6 to 0.9 and an underestimation of other CFs. In the winter, when the SW bias is negative, there
is a positive bias in CF. In summer we find an underestimation of CF, when the positive SW bias is particularly evident, which agrees with the previous studies' conclusions. Nevertheless, the overall overestimated CF and positive surface SW biases in the model indicate that the CF alone does not control the cloud radiative effect, but also properties such as cloud phase, cloud base height, and cloud geometrical or optical thickness are likely to play a significant role (Viúdez-Mora et al., 2015; Cesana and Storelvmo, 2017; Fiddes et al., 2022). In addition, cloud microphysics such as ice crystal shape and size distribution and direct
and indirect effect of aerosols could also have an effect on radiation biases (Bohren and Huffman, 2008; Kuma et al., 2020). Our results here are in agreement with the work done by Schuddeboom and McDonald (2021), which found overestimated low-level CF and reduced reflectivity of low-level cloud over the SO in CMIP6 models, highlighting the significance of correctly representing low-level clouds to simulating radiative balance over the SO. In the next section, we examine how CF influences the radiation and, as a result, the CRE.

**4.2   CF distribution with respect to different radiation biases**

Figure 7 shows the CF distribution divided into different radiation bias cases over the entire time series. Figure 7a shows where the SW bias is large and negative, represented by the $10^{th}$ percentile (smaller than -20 W m$^{-2}$). Figure 7b is where the SW bias is small, between the $30^{th}$ and $70^{th}$ percentile (within $\pm 10$ W m$^{-2}$), and Figure 7c is where the SW difference is large and positive, represented by the $90^{th}$ percentile (larger than 50 W m$^{-2}$). For LW cases on the right panels, the selection criterion is
identical in terms of percentiles to the SW cases, but with different thresholds as indicated.

In Figure 7a, showing the negative SW bias condition, an evident overestimation of CF in the model is found, particularly at higher frequencies of CFs above 0.8. For Figure 7b, the CF in the model conforms well to the observation with a 0.02 difference on average. When the SW bias corresponded to a strong positive value (Figure 7c), the model largely underestimated the CF by overestimating the frequency of low CF (smaller than 0.4) and underestimating the frequency of high CF (larger than 0.8).
For panels on the right depicting the conditions restricted by the LW bias, Figure 7d shows that when there was a strong negative LW bias, the CF simulated by the model was significantly lower than the observations, influenced by a large underestimation of CF frequency of above 0.8. When the LW bias is relatively low (Figure 7e), simulated CF was comparable to the observation, with a bias of 0.03 on average. During large and positive LW bias conditions (Figure 7f), the model overestimated the CF by simulating too much high CF.

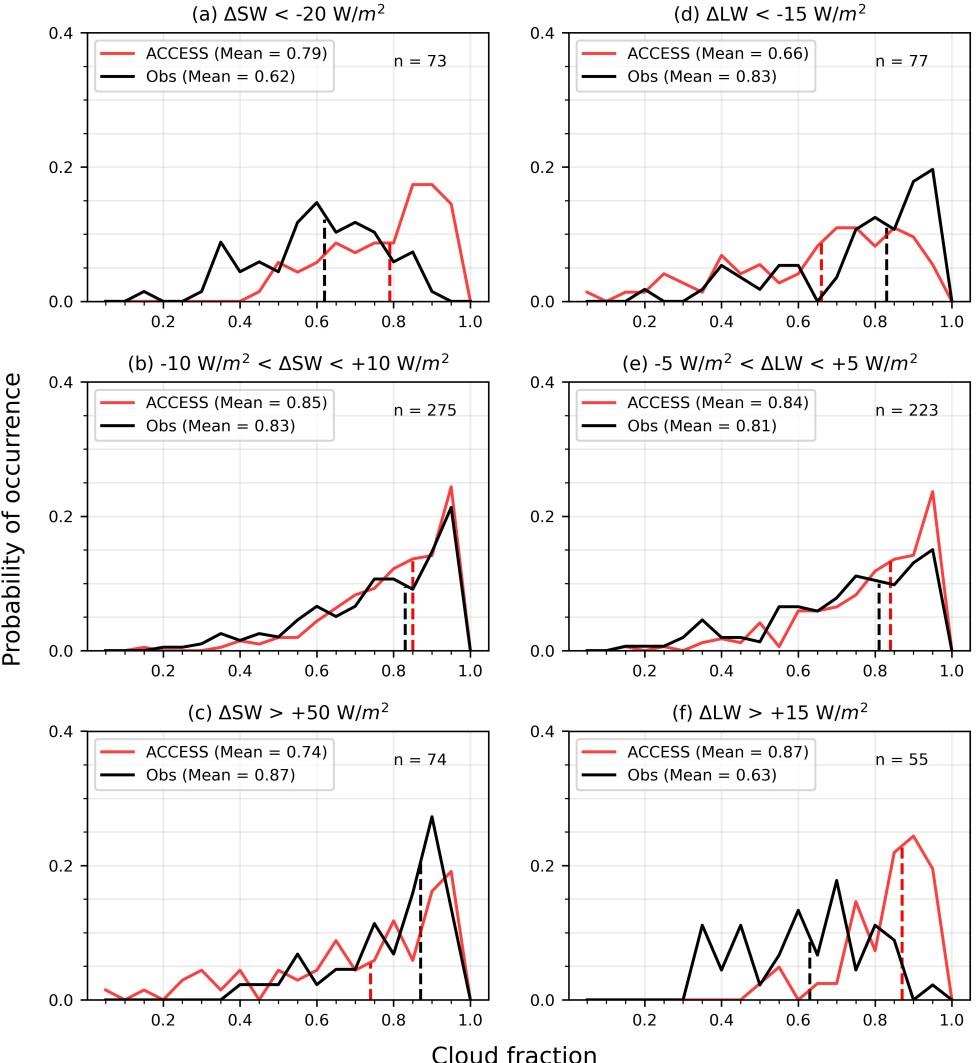

**Figure 7.** Daily averaged CF distributions from ACCESS-AM2 model (red) and observation (black), restricted to cases where downwelling surface (a) $\Delta$SW (ACCESS-AM2 – observation) < -20 W m$^{-2}$, (b) -10 W m$^{-2}$ < $\Delta$SW < 10 W m$^{-2}$, (c) $\Delta$SW > 50 W m$^{-2}$, (d) $\Delta$LW (ACCESS-AM2 – observation) < -15 W m$^{-2}$, (e) -5 W m$^{-2}$ < $\Delta$LW < -5 W m$^{-2}$, and (f) $\Delta$LW > 15 W m$^{-2}$. The amount of data in each case is annotated on the top right of the panel. The vertical dashed lines indicate the mean of the datasets corresponding to their colour. Averages are shown in the bracket after the legend.

The analysis above demonstrates that the radiative biases at the extremes can be associated with biases in CF, where SW radiation is underestimated with a greater CF, and is overestimated with a lower CF. LW radiation appears to consistently respond by following the opposite mechanism. For the majority of the time however, when radiative biases are small, we find that the model performs surprisingly well with respect to CF. This result suggests more attention should be paid to instances when the radiative biases are strongly positive or negative, to understand what type of cloud conditions are contributing to

these biases (e.g. cloud top pressure, cloud optical and geometrical depth). Moreover, it is worth noting that limitations exist in comparing CF derived from a camera versus a model as they have different spatial coverage. While much of that is smoothed out by taking daily averages, the statistics could still be affected.

## 5   Cloud frequency of occurrence

The ALCF product was operated using the hourly data from ACCESS-AM2 model and ceilometer data from September 2017

to February 2018. Figures 8 and 9 show the likelihood of cloud occurrence with height over this period of time. The CF in the figure caption denotes the overall CF collected by the ceilometer over the six-month period as calculated by ALCF, which might include both fog and precipitation. Averaged CFO and CF of 10 subcolumns are chosen to represent the model's statistics. The results in this section can be seasonally biased as the date range is not an integer number of years. The different magnitudes of overestimation or underestimation of CF compared with previous results in Section 4.2 may be attributed to the different time

period of the data. Additionally, different viewing geometry (the ceilometer only sees directly overhead, whereas the camera sees a much larger part of the sky), detection thresholds, any errors in cloud detection (incomplete overlap near the surface in the ceilometer), or precipitation misidentified as cloud could cause the differences in CF measured by the cloud camera and ceilometer.

    Figure 8 shows the histogram of CFO, measured by the ceilometer at Macquarie Island, versus height above the mean sea

level. It demonstrates an observed predominant low-level cloud between the surface and 2 km height. Cloud occurrence was found 35% of the time near the lowest level (around 50 m) provided by ALCF. As the altitude increases, cloud occurrence declines rapidly, reaching close to zero above 6 km, which could be partially due to the backscatter attenuation caused by low-level clouds (McErlich et al., 2021). In comparison, the model has a large underestimation of cloud occurrence at the lowest level where only about 13% of the cloud occurrence is detected near the surface. It reaches a peak at around 500 m and then

gradually diminishes with height. In contrast to the ceilometer, the model detected small cloud occurrences above 6 km. The overall CF for this period (spring and summer) observed by the ceilometer was 89%, which the model underestimated by 2%.

    Ceilometers determine the CBH from the backscatter profile, and higher clouds can be obscured by optically-thick low-level water clouds, rendering high-level clouds invisible in the profile (Klekociuk et al., 2020; Kuma et al., 2021; McErlich et al., 2021). The limitation of the ceilometer is likely the explanation for clouds above 6 km to be unaccounted for in these

observations. Multilayer cloud occurrence of 19.5% was obtained by Protat et al. (2017) within a span of 10 days between latitudes of 43°S to 48°S. Klekociuk et al. (2020) found a 26% occurrence of multilayer cloud during a two-month campaign from latitudes 44.7°S to 67°S. By examining these previous observations, we can have an approximation of the frequency at

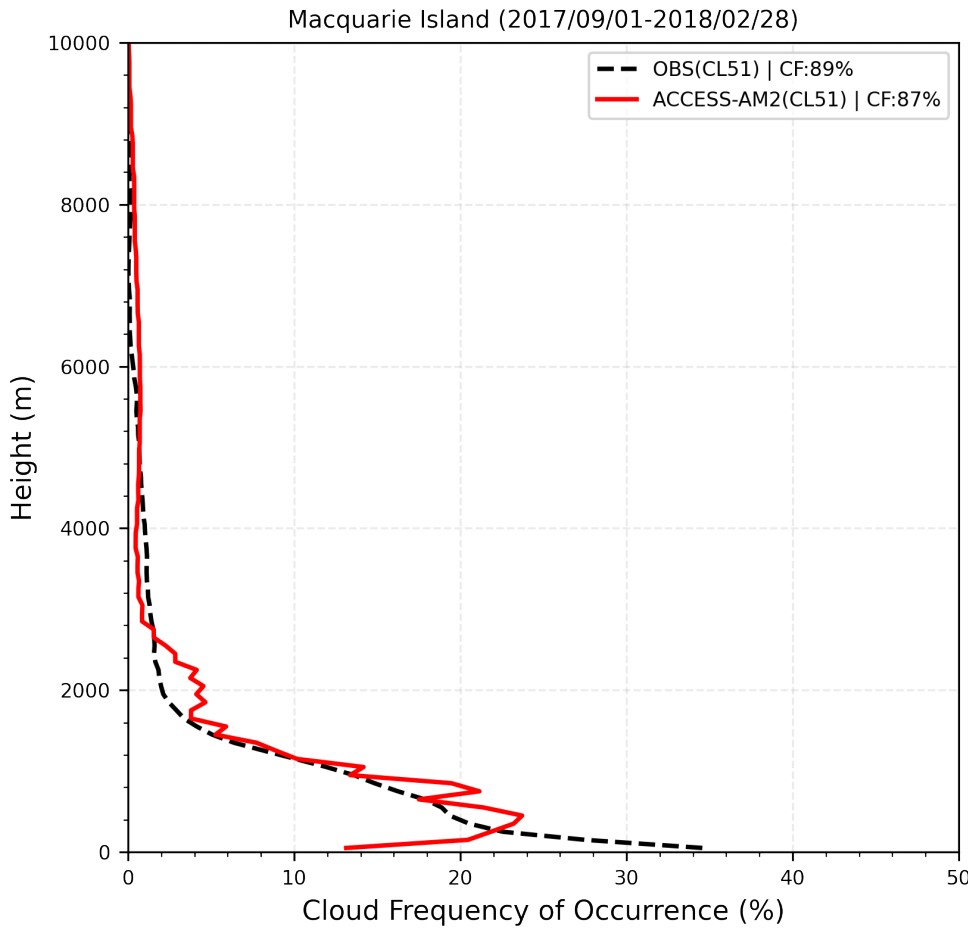

**Figure 8.** CFO histogram against height above the mean sea level observed at Macquarie Island (black) and simulated by the lidar simulator based on atmospheric fields for ACCESS-AM2 model (red), from September 2017 to February 2018. The total CF is shown in the legend. The height of this plot is limited to 10 km as there was no significant amount of cloud detected above this level.

which the ceilometer experiences the limitation of high cloud obscurity over the SO. As the ACCESS-AM2 outputs are passed through the ALCF - the same limitation applies to the modelled data. Therefore, we can fairly compare the characteristics of low-level clouds between the ceilometer and model. When using the ALCF to replicate what the model 'observes' if it were a ceilometer, Figure 8 highlights that the ACCESS-AM2 model tends to underestimate the total CF and can only partially reproduce low-level cloud below 300 m while slightly overestimating the cloud occurrence between 300 m and 3 km, with the largest overestimation of 5% observed at 800 m. In Fiddes et al. (2022) it was found that the ACCESS-AM2 model, when compared to COSP satellite products, underestimates low clouds at the expense of mid-level clouds. The NWP version of the ACCESS model underestimated the low-level cloud occurrence below 1.5 km and largely overestimated the frequency of multilayer cloud, consistent with the excess surface SW radiation in the model (Protat et al., 2017). Different from prior studies that demonstrated an overall underestimation of low-level clouds below 2 km in the ACCESS model, our findings suggested a specific underestimation of low-level clouds below 300 m and overestimation for other low-level clouds above this altitude. The general underestimation of total CF conforms to previous studies that used the ALCF to assess the performance of several climate models and reanalysis products (Kuma et al., 2020, 2021).

Nevertheless, it is crucial to note that limitations exist in ALCF for reproducing CFO. As mentioned in Section 2.6, ALCF doesn't identify precipitation, which could be classified as cloud in the ceilometer while ignored in the model (Kuma et al., 2021). This may cause an overestimation of CFO near the surface in the ceilometer and potentially amplify the underestimation of low-level CFO in the model. Upon visually inspecting the time series of ceilometer backscatter profiles, certain layers beneath stratocumulus clouds at around 500 m are identified as clouds, potentially consisting of drizzle, snow, or fog. Tansey et al. (2022) has reported an occurrence of 34% and 19% of drizzle in 2016-17 spring and summer at Macquarie Island. Moreover, Stanford et al. (2023) found that ceilometer observations on Macquarie Island were obscured 18 % of the time because of fog, which is also likely to influence the CFO near the surface. Hence, low-level CFO below 500 m should be interpreted cautiously as it could be influenced by the combination of precipitation and fog. Further research that combines lidar/ceilometer with precipitation measurements will be beneficial to the model evaluation. Moreover, more sophisticated algorithms to classify precipitation, fog, and aerosol are suggested to be developed within ALCF.

To further explore the relationships of cloud occurrence with the radiative biases, we next evaluate the cloud occurrence under different radiation bias conditions, as described in the previous section. Figure 9 shows the histograms of cloud occurrence and total CF in different radiation bias conditions, noting that we are not showing the cloud occurrence profiles under neutral bias conditions, as the model performed very similarly to the mean shown in Figure 8 for both SW and LW conditions.

When SW biases are large and negative, as per the $10^{th}$ percentile, the model generates 16% higher total CF than the observations and simulates the same cloud occurrence of 8% near the surface (Figure 9a). There is a persistent overestimation of cloud occurrence between surface and 2.5 km in the model, compensated by the underestimation of cloud occurrence between 2.5 km and 6 km. The observed CFO in these conditions is smaller than that observed under average conditions at lower levels, as seen in Figure 8 and replotted in Figure 9 in lighter colours. The model however, sees an increase of CFO above 1 km altitude. When the SW bias is large and positive (Figure 9b), as per the $90^{th}$ percentile, the model underestimates the cloud occurrence below 1.5 km, but up to 32% near the surface. The model's low-level cloud occurrence below 1 km decreased from

the mean (Figure 8), while the observed low-level CFO has clearly increased. The total CF simulated by the model was lower than observed by 15% in these cases.

Figure 9c, in which cases of large negative LW biases were selected by the $10^{th}$ percentile, demonstrates a strong under-estimation by the model of cloud occurrence below 1 km and a significant overestimation between 1.5 km and 3 km. The ALCF model output overestimates clouds above 4 km, possibly offset by the scarce simulated low-level clouds. The model output is also clearly different to that of the mean modelled occurrence profile, where low-level cloud below 1 km occurred less frequently. This is in contrast to the observed profile, where we see more frequent cloud occurrences between surface and

approximately 1.5 km. The total modelled CF was lower than the observed values by 31%. When the LW bias is large and positive (Figure 9d), as per the $90^{th}$ percentile, the model tends to overestimate the cloud occurrence below around 3 km and significantly overestimate the total CF by 41%. The observed low-level CFO is lower than the average conditions in these cases, with a similar profile to that of the underestimated SW bias profile (Figure 9a). Below 500 m, the modelled cloud occurrence is lower than the average, but greater between this level and 2 km.

Combining the cloud occurrence and CF over different bias conditions above, excessive downwelling surface SW radiation in the model was associated with lower low-level cloud occurrence and lower CF, which aligns with our expectations. The mid- and high-level clouds can be simulated by the model even though the model underestimates low-level clouds. The minor changes in the modelled cloud occurrence profiles between both SW conditions and the mean demonstrate the models inability to capture a diverse range of cloud types, as found in Fiddes et al. (2022). However, this relative consistency in vertical cloud

profile appears to be less apparent when considering the LW conditions. For the LW bias conditions, larger differences in the modelled low-level cloud occurrence profiles are observed, when compared to the mean. This may suggest that the CFO of low-level clouds has a larger controlling factor over the LW biases than that of the SW biases.

Despite the large influence of cloud macrophysical characteristics such as CF and CFO on cloud radiative properties, it is essential to acknowledge the crucial role played by cloud microphysical properties such as cloud phase, cloud droplet

number concentration, and cloud effective radius. Vergara-Temprado et al. (2018) suggested the significance of incorporating the spatial and temporal variations of ice nucleating particle (INP) concentrations in cloud microphysics scheme. More realistic INP distributions and cloud microphysical properties are crucial to accurately simulate cloud phase, cloud reflectance and thus radiation (Tan and Storelvmo, 2016; Furtado and Field, 2017). Gettelman et al. (2020) compared cloud microphysics in a nudged global climate model (the Community Atmosphere Model, CAM) with aircraft observations (the Southern Ocean

CLouds, Radiation, Aerosol, Transport Experimental Study, SOCRATES) collected over the SO. An improved simulation of SW CRE was shown by implementing a revised autoconversion scheme that reduces both liquid and ice water path but increases cloud fraction and effective radius, maintaining more supercooled liquid water. Nevertheless, the model still fell short of matching the droplet numbers observed in aircraft measurements, which suggests that higher concentrations of cloud condensation nuclei (CCN) and greater droplet numbers may be required to achieve better agreement (Gettelman et al., 2020).

In light of these preceding studies, a more detailed understanding of cloud macro- and microphysical properties is necessary to correctly simulate the radiation balance in climate models.

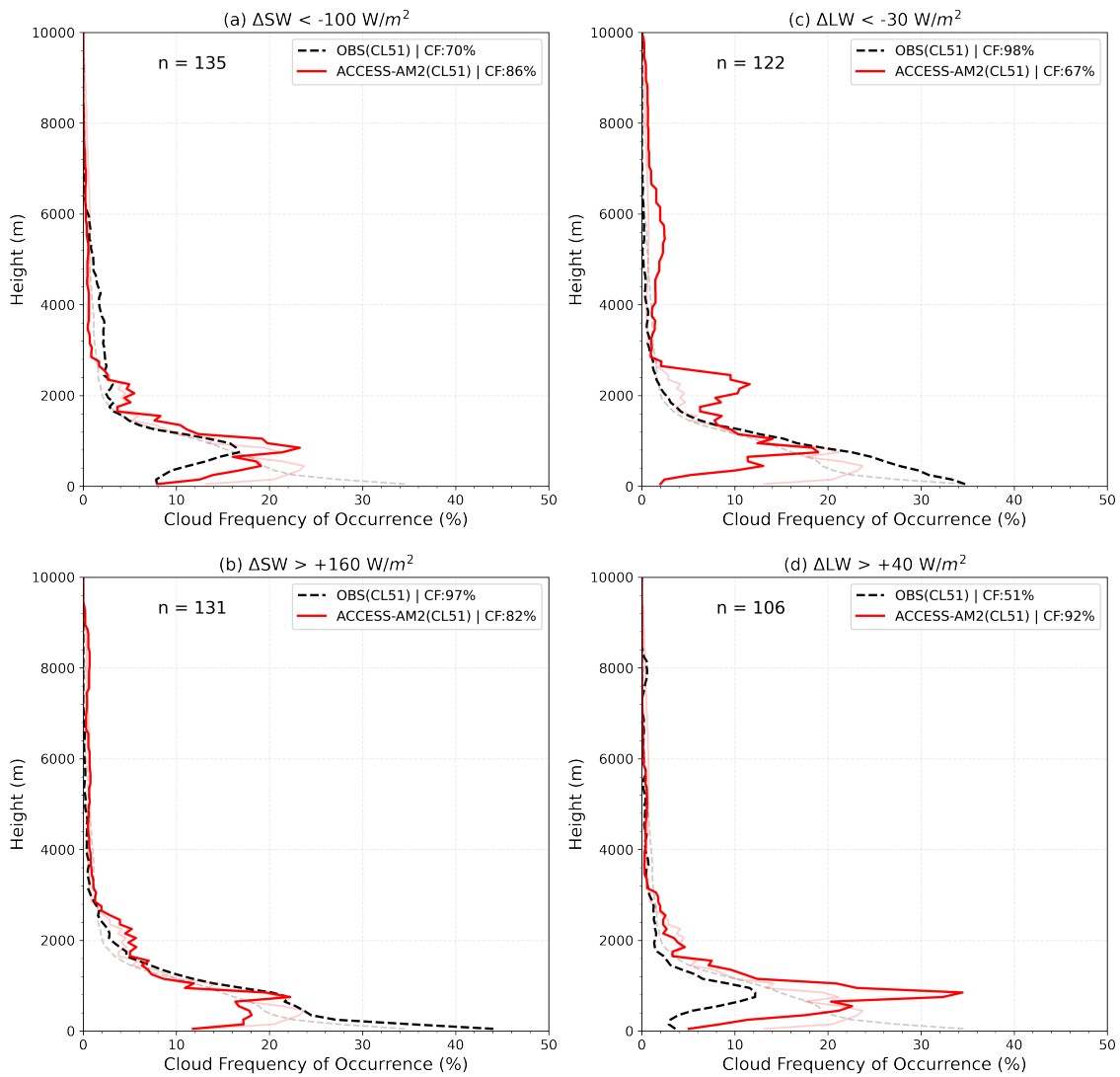

**Figure 9.** Same as Figure 8 but restricted with different bias conditions, where (a) $\Delta$SW (ACCESS-AM2 – observation) < -100 W m$^{-2}$, (b) $\Delta$SW > 160 W m$^{-2}$, (c) $\Delta$LW (ACCESS-AM2 – observation) < -30 W m$^{-2}$, and (d) $\Delta$LW > 40 W m$^{-2}$. The lighter shaded lines indicate the total cloud occurrences as in Figure 8.

# 6 Conclusions

In this work, we provided an evaluation of radiation fluxes, CRE, CF and cloud occurrence for the ACCESS-AM2 model using surface-based observations between April 2016 and March 2018 at Macquarie Island. In addition, we evaluated the radiation fluxes and CRE for the CERES SYN1° surface-based product over the same period. Moreover, we used the newly developed lidar simulator, ALCF, to quantify the relationship between cloud occurrence and radiation biases in the model.

For the ACCESS-AM2 model, there was an overestimation of $+9.5 \pm 33.5 \,\mathrm{W\,m^{-2}}$ for downwelling surface SW radiation fluxes and an underestimation of $-2.3 \pm 13.5 \,\mathrm{W\,m^{-2}}$ for LW radiation fluxes in all-sky conditions. The SW bias was more pronounced in spring and summer on account of reduced low-level CF in the model, as well as strong solar radiation during these seasons. The slight LW bias suggests a good performance of the model in simulating LW radiation on average. Compared to ERA5, a small underestimation of $-2.3 \pm 3.7 \,\mathrm{W\,m^{-2}}$ for SWcs radiation and a significant overestimation of $+4.5 \pm 5.3 \,\mathrm{W\,m^{-2}}$ for LWcs were found in the model, despite also being nudged to ERA5. The combination of radiation biases in all-sky and clear-sky conditions contribute to an overestimation of $+4.8 \pm 28.0 \,\mathrm{W\,m^{-2}}$ for total CRE in the model, dominated by the SW CRE bias of $+11.2 \pm 31.1 \,\mathrm{W\,m^{-2}}$. The total CRE bias was more pronounced in summer, which can be attributed to the SW CRE bias. In winter, the LW CRE bias contributes most of the total bias. For the CERES product, there was an overestimation of $+8.0 \pm 18.0 \,\mathrm{W\,m^{-2}}$ for SW radiation fluxes and an underestimation of $-12.1 \pm 12.2 \,\mathrm{W\,m^{-2}}$ for LW radiation fluxes in all-sky conditions, with the SW bias dominating in summer and LW bias dominating in winter. These results agree with Hinkelman and Marchand (2020), who showed a poor simulation of low-level CBH at night contributing to the LW bias in the satellite measurements. For clear-sky conditions, the SWcs is well captured by CERES, while the biases of LWcs are significant (despite also using ERA5 to inform the radiative model) and very similar to the model. We speculate that temperature and humidity representation play an important role in causing the LWcs bias in CERES, and suggest that further research should be conducted to evaluate clear-sky radiation properties in CERES and ERA5.

The average CF distribution simulated by the model is comparable with the observations with a bias of 0.01. However, this is caused by an underestimated frequency of CF between 0.2 and 0.6 and an overestimated frequency of CF above 0.6. Unlike prior summer-focused studies that found an underestimation of CF in the model, in this study we found an overestimation of mean CF across the year (0.01) with the exception of a slight underestimation in summer (0.02). This highlights the need for greater model evaluation throughout the seasons as the summer biases may not be representative throughout the year. When restricting CF to different radiation bias conditions, an overestimation of surface SW radiation was associated with an underestimation of CF, and an underestimation of surface SW radiation was associated with overestimation of CF. The opposite was found for LW cases.

By using ALCF we can compare ceilometer data with the model data directly. Overall, the results highlighted an underestimation of low-level cloud occurrence below $300 \,\mathrm{m}$ and total CF, which may be biased due to the presence of precipitation and fog. When evaluating cloud occurrence under different radiation bias conditions, it was demonstrated that an overestimation of SW radiation is associated with an underestimation of low-level cloud occurrence and CF, and negative SW radiation biases were associated with overestimated low-level cloud occurrence. For different LW bias conditions, the results are opposite to the

SW bias conditions, as expected. An overestimation of LW radiation is associated with an overestimation of low-level cloud occurrence below 3 km, while the model severely underestimates low-level cloud occurrence for negative LW bias conditions. We suggest that the larger differences in the modelled low-level cloud occurrences between the LW conditions demonstrates the greater dependence on low-level CFO of the LW biases than SW biases.

Aside from the erroneous cloud representation in the model, radiative biases could also arise from data collection and processing, which must be considered. For example, observed missing data on specific days have been ignored. The simulated clear-sky radiations are based on the modelled ERA5 product, which will include inaccuracies on clear-sky radiation estimation due to inaccurate temperature and humidity profiles. The calculation of all-sky CF is influenced by the presence of sun in the cloud images. This results in a saturation of a portion of the fisheye image, resulting in uncertainty in the estimated CF.

Additionally, the limitation of the ceilometer (in both the observations and the ALCF derived product) in detecting high-level clouds adds difficulty for complete model comparison. For the ACCESS-AM2 and CERES, the Macquarie Island location is interpolated from coarse resolution grid-boxes, which will also bring about some unavoidable biases.

Overall, this study reinforces the finding of excess downwelling surface SW radiation in the ACCESS-AM2 model. The significant bias of surface radiation fluxes in the SO in CERES, which can lead to an underestimation of model bias, indicates

the requirement to also continually evaluate satellite products using ground-based observations. We also highlight the need to investigate what an accurate representation of clear-sky is in the SO, given the difficulty in validating current clear-sky models due to consistent cloudiness and its necessity in calculating the CRE. Moreover, this work confirms that the CF and cloud occurrence have a large impact on the surface radiation, though with differing importance for the SW and LW biases. We suggest that this demonstrates both the lack of diversity of clouds represented by the model, as suggested by Fiddes et al.

(2022) and also that other cloud microphysical properties, such as cloud phase, cloud effective radius, and cloud droplet size distribution may be more important than the vertical profile for the SW biases. We emphasize that the correct representation of supercooled liquid water over the SO is important for modelling the radiation in the region, as inadequate supercooled liquid water content will cause less reflectivity of clouds and result in positive downwelling surface SW biases (Luo et al., 2016; Vergara-Temprado et al., 2018; Gettelman et al., 2020).

For future studies, further evaluation of the climate models at more locations over the SO is suggested to comprehensively investigate the radiation biases over this region. In the latest Guyot et al. (2022) study, ALCF can now be used to detect the cloud phase, enabling future studies to address the role that cloud phase plays in influencing the radiation biases by further utilizing this tool. In addition, satellite products which showed non-negligible biases require further evaluation and development in surface radiation retrievals since this is still the primary tool in use for evaluating the model.

*Data availability.* The ARM radiometers data collected during MICRE are available via the ARM data archive (https://adc.arm.gov/). The AAD radiometer and all-sky cloud camera data are available at the Australian Antarctic Data Centre (AADC) (https://data.aad.gov. au/metadata/AAS_4292_Macquarie_Island_Radiometer_and_AllSkyCam_Data_2016-2018). The University of Canterbury's Vaisala CL51 ceilometer data are available at AADC (https://doi.org/10.26179/5d91835e2ccc3). The ACCESS-AM2 model data for this project are avail-

able at https://doi.org/10.5281/zenodo.6004062 (Fiddes et al., 2022). The satellite data (CERES SYN1deg) are available via the CERES data products (https://ceres.larc.nasa.gov/data/). The ERA5 data are available via the Copernicus data portal (https://cds.climate.copernicus.eu/). The ALCF is open-source and available at https://alcf.peterkuma.net and on Zenodo at https://zenodo.org/record/4411633.

## Appendix A: Validation of the measurements of AAD's radiometers against the colocated ARM's radiometers

Here we compare radiation measurements derived from radiometers deployed by AAD against co-located radiometers deployed by ARM (McFarquhar et al., 2021), to validate both data sets. Seasonal comparisons were made for SW radiation (Figure A1, left) from 4-April-2016 to 6-March-2018 and LW radiation (Figure A1, right) from 15-August-2016 to 6-March-2018. The linear regression coefficients range from 0.94 to 1.01 with no obvious seasonal differences, which depicts a good consistency between two datasets. Additionally, hourly comparisons were also plotted for two datasets with their 95-percent confidence intervals (Figure A2). Both datasets appear to agree and are for the majority of the time within the 95% confidence intervals. This analysis suggests there is little meaningful difference between the two co-located instruments giving us confidence in our results.

## Appendix B: Diurnal cycle and bias of total CRE, $CRE_{SW}$ and $CRE_{LW}$

In this section the CRE biases over the diurnal cycle were explored. We now consider only the period of September 2017 to February 2018 where hourly instantaneous output was available from the ACCESS-AM2 model. Figure B1 shows the diurnal cycle of total CRE, $CRE_{SW}$, $CRE_{LW}$ (top) and the associated biases compared with ground-based observations (bottom).

For ACCESS-AM2 total CRE (Figure B1a,d), small negative biases are found during the nighttime due to the lack of incoming solar radiation, contributed by the biases in $CRE_{LW}$. The highest difference for total CRE occurs around 1 a.m. UTC (approximately local solar noon), when the difference is roughly +57 $\mathrm{W\,m^{-2}}$. $CRE_{SW}$ has a similar diurnal cycle to the total CRE's, with the bias peaking at the same time as total CRE's. Throughout the day, the $CRE_{LW}$ is comparable with surface observations and shows no diurnal variability (see y-axis scale in Figure B1c). CERES $CRE_{SW}$ bias exhibits similarities to ACCESS-AM2, while it is larger during specific periods, such as 4 a.m. UTC, 19-21 p.m. UTC (Figure B1e). Different to ACCESS-AM2, CERES $CRE_{LW}$ has notable negative biases at night local time (7 a.m. - 18 p.m. UTC), with biases ranging from -20 to -15 $\mathrm{W\,m^{-2}}$. The significant underestimation of $CRE_{LW}$ in CERES, as highlighted in Hinkelman and Marchand (2020), is attributed to incorrect cloud base height during local nighttime periods (Figure B1c,f).

*Author contributions.* Z. Pei completed the data processing, analysis and manuscript writing. S. Fiddes provided the model simulations. J. French provided data processing for raw observational data. P. Kuma contributed to the operation of ALCF. S. Alexander and A. McDonald provided the observational data at Macquarie Island. Z. Pei, S. Fiddes, S. Alexander, J. French, and M. Mallet contributed to the project planning and data processing. All authors contributed to the revisions of this paper.

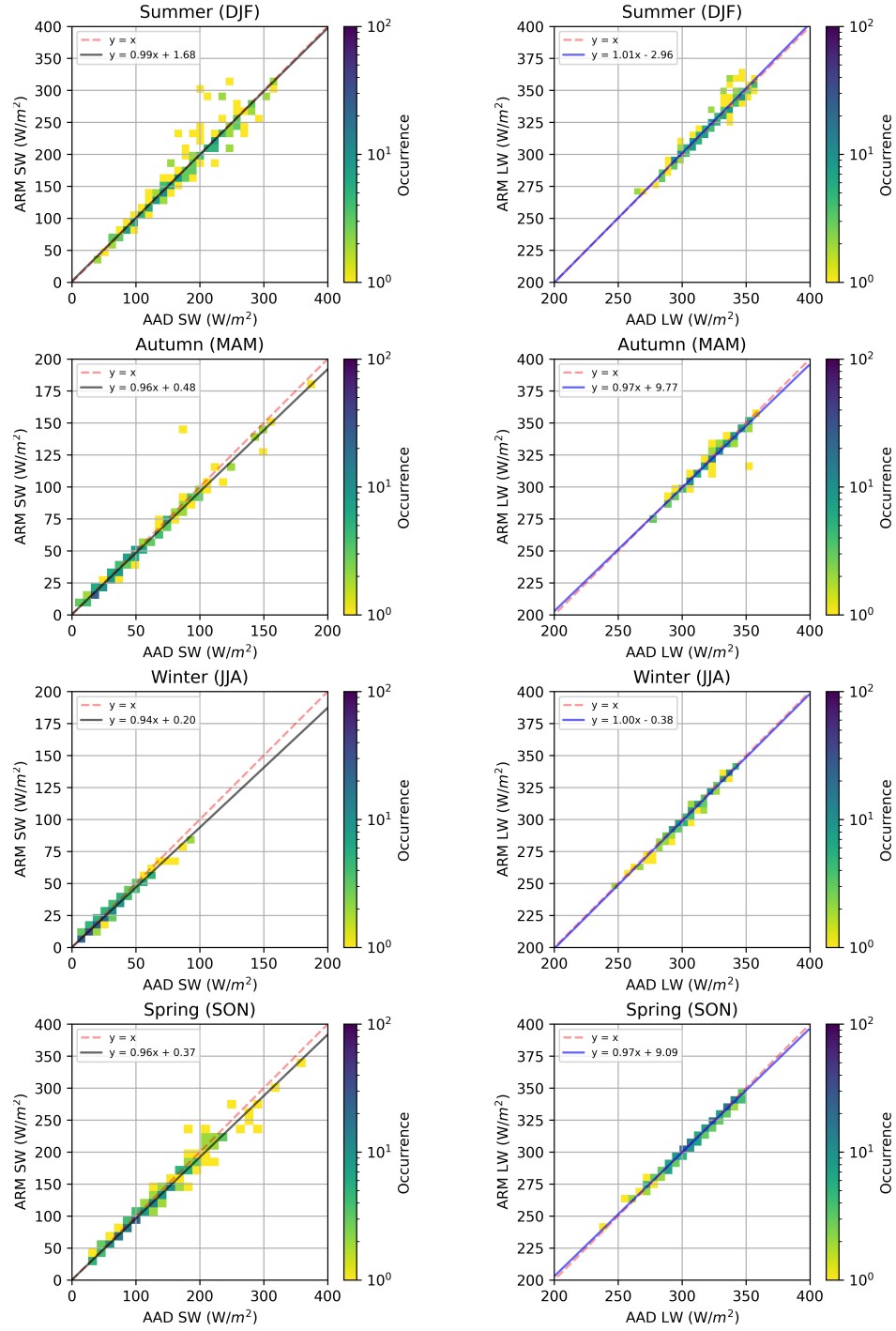

**Figure A1.** The seasonal comparisons of radiometers between AAD and ARM for 4-April-2016 to 6-March-2018 (pyranometer) and for 15-August-2016 to 6-March-2018 (pyrgeometer). Radiation data are averaged daily, and linear fit parameters are detailed.

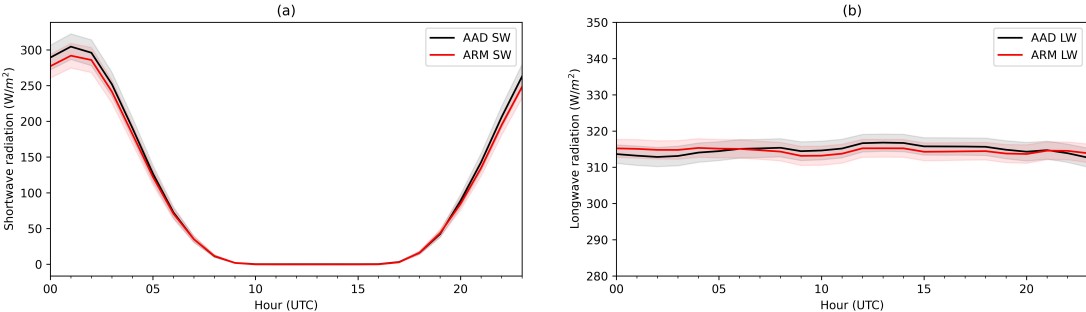

**Figure A2.** Hourly plots of SW (a) and LW (b) radiation data for AAD and ARM measurements, with the shaded areas showing the 95-percent confidence intervals.

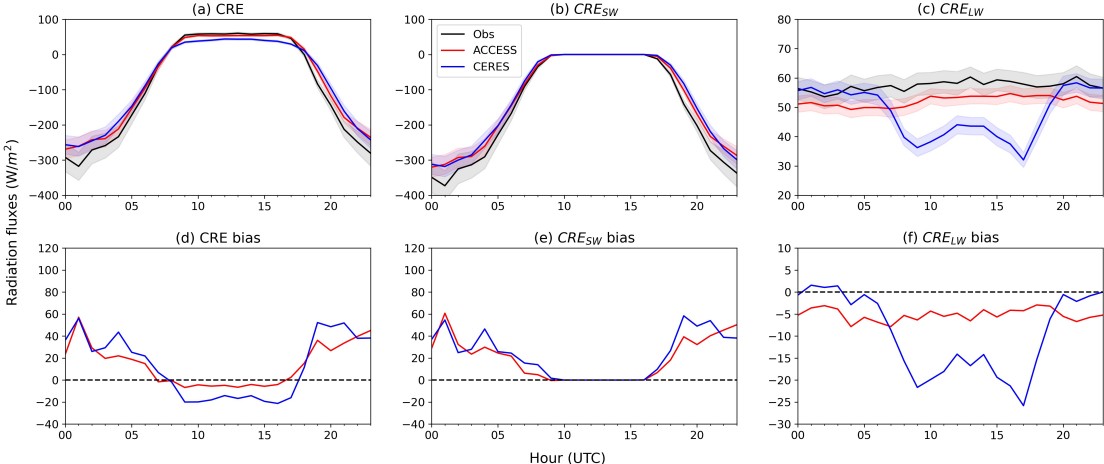

**Figure B1.** Diurnal cycle of the ACCESS-AM2, CERES and observational CRE (a), $CRE_{SW}$ (b) and $CRE_{LW}$ (c) in the above, associated with the biases (ACCESS-AM2 – Observation; CERES - Observation) in the below (d) (e) (f), based on data from September 2017 to February 2018. The shaded areas of the panel above represent the 95-percent confidence interval of the value.

*Competing interests.* The authors have no competing interests to declare.

*Acknowledgements.* This project received grant funding from the Australian Government as part of the Antarctic Science Collaboration
Initiative program, under the Australian Antarctic Program Partnership, ASCI000002. This research was undertaken with the assistance of
resources and services from the National Computational Infrastructure (projects jk72, hh5 and rt52), which is supported by the Australian
Government. Technical and logistical support for the deployment to Macquarie Island were provided by the Australian Antarctic Division
through Australian Antarctic Science Project 4292, and we thank Andrew Klekociuk, Peter de Vries, Terry Egan, Nick Cartwright and Ken
Barrett for all of their assistance. Z.P. and S.F. would like to thank the ARC Centre of Excellence for Climate Extremes computer modelling
support team for their maintenance of virtual environments and code/model support. P.K. would like to thank the nextGEMS project funded
by the European Union Horizon 2020 grant number 101003470. The authors would like to acknowledge the teams at DOE ARM, NASA
CERES and ECMWF for making the data used in this work available.

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
