# Peer review of "Assessing the cloud radiative bias at Macquarie Island in the ACCESS-AM2 model"

_EGUsphere, 2023_

## Referee Comment (RC1)

Pei et al. compare surface observations of downwelling shortwave and longwave radiation at Macquarie Island to the ACCESS-AM2 model and CERES product. They then analyze radiative biases as a function of cloud fraction and cloud occurrence, using the ALCF simulator (relying on hourly output from the model) and an all-sky cloud camera for the observational component. Overall, the authors clearly relate respective CRE biases to the identified shortcomings of both ACCESS-AM2 & CERES, in comparison to the MICRE observations. They establish a strong relationship between cloud fraction and SW/LW at the surface in both the model and observations. Interpretation of the role played by cloud occurrence (especially low clouds) is weaker, and I recommend that the authors provide more background information on the role of cloud microphysics (beyond cloud phase). This may include a brief survey of studies of Southern Ocean cloud properties from surface campaigns and in-situ aircraft measurements (e.g. during SOCRATES). I also recommend that the authors report the fraction of (and reasons for) missing/bad data for *all* observations (and whether you expect these to introduce any sampling bias), and more detail on the limitations of using ERA5 for clear-sky radiation estimates. I recommend that this paper be published after minor changes.
* * *
General comments

1) In the introduction section second paragraph, more could be said about the sparseness of surface-based SO ceilometer observations (cloud occurrence frequency and cloud boundaries) and sources of uncertainty within the ALCF data. You state that this is a recently developed tool. Include a brief description of the ALCF here (with more detail in methods).

2) Regarding uncertainty, data issues and missing data fractions for all surface-based measurements: You provide a good description of this for the radiometer. For the MICRE ceilometer, can you provide a % of time that you do not have good data (missing a successful cloud base retrieval) because of fog, snow on the detector or other reasons for missing data? How does fog and attenuation affect your CFO profiles? (See also "general comment" #4 below.)
In reference to the all-sky cloud camera, can you report the uncertainty and bad/missing data %?

3) In methods section 2.3.2, you state that you're able to contain clear-sky radiative biases in equation 4 by relying on ERA5, which ACCESS-AM2 & CERES also rely on. But this re-emerges later in the paper as a driver of uncertainty in the interpretation of results. Are you aware of any literature worth mentioning (whether early in the paper or in the discussion section) that provides more information on ERA5 radiative biases over the region? I scanned briefly and found one very recent article, for example:

Mallet, M. D., S. P. Alexander, A. Protat, and S. L. Fiddes, 2023: Reducing Southern Ocean shortwave radiation errors in the ERA5 reanalysis with machine learning and 25 years of surface observations. Artif. Intell. Earth Syst., https://doi.org/10.1175/AIES-D-22-0044.1

4) Figures 8 & 9 are of concern because of the very large CFO near the surface. Describe in detail how you got this profile and how you calculate cloud fractions at each height. Especially near the surface, >50% cloud occurrence in the lowest 50 meters seems improbable. If your approach was to define a minimum threshold on the ceilometer backscatter, you should discuss (and potentially rethink) this.

5) In the interpretation of results in section 5, I think you should discuss other possible sources of radiative biases here (e.g. differing cloud macro and microphysical properties). It is not sufficient to state only that the low-level CFO was less negative for negative SW biases, and provide no additional reasoning. If two opposing extremes of SW biases are both happening with underestimated low-level CFOs, additional cloud properties must play a role here.
Lines 502-505 start to address this. I think the discussion (and the paper overall) would be strengthened by referencing additional studies of cloud radiative effects as a function of macro and microphysics.

6) You have two years' worth of seasonal cycles, and yet you do not discuss the differences in the two years. It would be interesting to see how much downwelling radiation, cloud fraction and vertical occurrence frequencies varied in each season during 2016-2017 as compared to 2017-2018.

7) Appendix B: What about diurnal errors in the CERES SYN product?
* * *
Line-specific comments

1) Line 50-52:  You should also acknowledge here that discrepancies even exist between surface and satellite observations, due to limitations of satellite near-surface cloud retrievals.

2) Line 55-56:  I suggest connecting cloud feedbacks to the previous sentences; what cloud feedbacks are relevant to this study?

3) Line 62-63: Suggest rephrasing "remote atmospheric environment" to "the harsh atmospheric environment and lack of remote sites for measurements"

4) Line 99-101: This explanation makes it difficult to follow what the time resolution is. 1-minute means and standard deviations? I suggest stating this up front.

5) Line 104: I suggest writing out "microvolts"

6) Line 107: Please describe the calibration process in more detail. How did you get these coefficients? How do you determine sensitivities?

7) Line 119: Consider relating the 0.5°C to a % error in the measurements. Presumably this will have a very minimal effect on equation (1) since $T_b$ is in Kelvin, although it is raised to the fourth power.

8) Line 123: "*limited* clipped points"

9) Line 127-129: In my opinion this should be stated earlier in the section.

10) Line 133: Do you decrease 6-second sampling to 1-minute resolution to match up with other data? If so, what is your approach (such as using the median CBH, do you require a minimum detection threshold i.e. X columns out of 10 columns in the minute contain cloud base retrievals?)

11) Line 190: Explain why September-February were chosen to look at hourly simulations.

12) Line 212-214: Provide some comment on the input to the ALCF simulator.

13) Line 217: misspelled ceilometer

14) Line 218-219: What is the spacing of height bins? What is a column? E.g. a 1-minute vertical profile of 100-meter vertical bins extending up to 15km?

15) Line 233: "Generally consistent" - I suggest making this statement quantitative: ACCESS, CERES match observations to within X,Y in W/m^2, "as seen in table 1". You start to do this at line 250, but a comment on general consistency can be made earlier to prepare the reader for what constitutes a significant disagreement.

16) Line 237: Curiously, this underestimation seems more pronounced in the second winter (JJA 2017) when I look at Figure 2. Is this the case? If so, perhaps comment on this in the text?

17) Line 254-255 (and re: Figure 3 in general): Perhaps add a sentence here or in the figure caption explaining the violin plot spread at various radiative fluxes (vertical axis values), and what the bolded segment is meant to show on the middle line.

18) Line 256: I suggest a rewrite, "reaching -4 W/m^2 in Autumn, with smaller differences in all other seasons."

19) Line 275-276:  In more recent generations this "too few, too bright" issue has been modified -- perhaps talk about findings from Schuddeboom and McDonald (2021).  Are the cloud fraction differences discussed in agreement with this analysis?

Schuddeboom, A. J., & McDonald, A. J. (2021). The Southern Ocean radiative bias, cloud compensating errors, and equilibrium climate sensitivity in CMIP6 models. Journal of Geophysical Research: Atmospheres, 126, e2021JD035310. https://doi.org/10.1029/2021JD035310

20) LIne 289 You write that the biases found here (SW +8.0 ± 18.0Wm$^{-2}$, LW bias of -12.1 ± 12.2Wm$^{-2}$) are "NOT consistent" with the previous study by Hinkelman and Marchand (2020) (SW +10Wm$^{-2}$, LW -10Wm$^{-2}$). How so? The means would appear to be nearly the same, and well within the listed uncertainty?
In general, I don't understand the ± values given here, as they don't seem to match table #1.

21) Line 292:  Isn't the ARM pyranometer from Hinkelman & Marchand [2020] the one used for the comparisons in Figs. A1 & A2? One would think calibration offsets and local shadowing effects have only minor impacts since the comparisons in the appendix are in such good agreement.
Also earlier you said only 9 days have been removed from your timeseries. So are data gaps in the ARM pyranometer more frequent?

22) Line 296.   You write "Excellent alignment of SWcs radiation between the satellite and reanalysis is expected given the CERES product uses ERA5 to inform its radiative transfer algorithm." Perhaps start a new paragraph as this represents a change in topic.

23) Line 312-313:  I would argue from downwelling SW in Fig. 3d that the winter and spring clear-sky LW are somewhat lower, exhibiting a weak but discernible seasonal cycle. Suggest removing "clear-sky" or changing to "similar to the all-sky and, to a lesser degree, clear sky"

24) Line 315:  "Figure 5 *shows*"

25) Line 319:  "During winter, when total CRE is at the lowest value" – JJA is the season where the CRE is highest (the most positive value). And if you are talking about magnitudes only, it is lower in MAM than in JJA. I'm not sure what you're referring to here.

26) Line 325:  Suggested that you change "climate models" to "ACCESS-AM2"

27) Line 329:  "which *peaks*"

28) Caption of Figure 6 & line 341:  Use the plural, "*observations*"

29) Line 342:  Draw the reader's attention to specifically Fig. 6a to demonstrate the integrated effect discussed in this sentence.

30) Line 343:  Only Figure 6e is for spring.

31) Line 350:  "mid-level CF":  You might include the thresholds here (above 4 km / below 10 km or something).

32) Line 357:  "*particularly*"

33) Line 360:  If you state that microphysics plays a lesser role in cloud radiative effect than the macrophysics described in the previous sentence, you should back this up with references.  More references in general in lines 358-363 would be good.

34) Line 369:  "*thresholds*"

35) Line 374:  Period after "LW bias" should be a comma

36) Line 382:  "This *result*"

37) Line 388-389:  Over what number of hours are you calculating cloud fraction? How are you averaging and comparing these two different datasets with different resolutions?

38) Line 389:  "*Figures*"

39) Figure 8:  I recommend adding shading around the vertical cloud fraction profiles indicating e.g. standard error on the mean CF at each height bin.

40) Line 395:  "overleap" – you probably mean overlap

41) Line 404:  "summertime only" – The period is spring and summertime.

42) Line 404:  I suggest rephrasing to  "which the model underestimated by 6%."

43) Line 405-406:  Look for references on how often multi-layer clouds occur. This can give you some estimate of how frequently the ceilometer suffers this drawback of high cloud obscurity.

44) Line 408:  I'm not sure I follow this. Why would ACCESS-AM2 output passed through ALCF suffer the same backscatter attenuation limitation?

45) Line 422-425:  Perhaps include a range or estimate of the uncertainty on the 4% higher total CF. In any case, especially considering the missing low-cloud occurrence < 1km, this result is surprising.
Additionally, I cannot tell by looking at 9a that the red line yields 4% higher CFO than the black line, all vertical levels considered... But this could just be an effect of the vertical axis log scale.

46) Line 425:  I suggest changing "Figure 8" to "as seen in Figure 8 and replotted in Figure 9 in lighter colors"

47) Line 428:  "*model's*"

48) Line 433: I suggest changing "simulates" to "overestimates"

49) Line 439: Awkward and needs rewording. I suggest changing to "Below 500m, the modeled cloud occurrence is lower than the average,"

50) Line 444: "can be in the model"? Maybe reword this.

51) Line 446: "cloud phase" – Cloud microphysics in general are likely contributing to SW radiative biases, including cloud droplet size and number concentration, cloud optical depth.

52) Line 470-471: Given the dependence on ERA5, can you speculate on the source of this problem?

53) Line 472: "*observations*"

54) Line 472-473: Elaborate more on these compensating errors as a separate sentence.

55) Line 473: Remove the comma after "prior"

56) Line 489: "inappropriate" is a funny word choice; I suggest inadequate or erroneous.

57) Line 490-491: If data are missing during these periods, what makes you say there were outliers in the observations?

58) Line 491-492: "which will include inaccuracies" – Such as?

59) Line 508: "*by* further"

---

## Referee Comment (RC2)

**Overview & Summary:** This manuscript evaluated cloud radiative biases in the ACCESS-AM2 model and the CERES satellite product against ground-based observations during the MICRE campaign at Macquarie Island. The study used a suite of instrumentation to establish a positive bias in surface shortwave radiation in both the model and CERES, which is consistent with numerous past radiative flux modeling studies over the Southern Ocean. The authors also employed a new lidar simulator to constrain model output with observational instrument specifications, therefore allowing an evaluation of cloud attribution to the radiative biases. They found that excessive absorbed surface shortwave radiation is associated with too low of a cloud fraction. Overall, the manuscript is well-written, descriptive where needed and brief where appropriate, and the experiment is well-designed. The largest concern is the lack of discussion of the potential role that cloud microphysics have in the discussed biases. While I understand this is not a microphysics study, there are long-standing microphysical biases in GCMs that are suggested to be relevant to simulating Southern Ocean clouds (e.g., prevalence and maintenance of supercooled liquid) that could be discussed in more detail as potential caveats. Otherwise, I have provided some general comments and line-specific comments below, none of which should require too much extra effort to delay publication. I suggest this manuscript be accepted for publication after these minor revisions are addressed.

**General Comments:**

- In the second paragraph of the Introduction when you introduce the ALCF, I think some expansion is needed for unfamiliar readers of how this framework operates. Typically, simulators are thought of as being applied to model data, not on observational data, so this can be confusing and deserves a little more explanation. Based on your description in Section 2.6, it seems ALCF is used on the observational ceilometer dataset as more of a means of calibration rather than "simulation".

- In the fourth paragraph of the introduction, I think something should be said about satellite limitations in observing low-level clouds over the SO, which is a very strong motivation for evaluating ground-based observations. As one example, Tansey et al. (2022; https://doi.org/10.1029/2021JD035370) looked at surface precipitation measurements during MICRE in comparison with CloudSat, with a few notable differences based on satellite instrument sensitivities and algorithm structure.

- You discuss how observed cloud fraction is defined by the all-sky cloud camera (Section 2.2.3) and then state it is used to evaluate CF biases on line 337 in Section 4, but it's not entirely clear how CF is being defined by ACCESS-AM2. Is it the prognostic value? Is it the CF computed by ALCF? It would be good to mention this at the beginning of Section 4.

- Section 2.1 and other parts of manuscript—I wouldn't call this in-situ observations, since many of what is included are more commonly thought of as passive remote sensing instruments. Perhaps change to "ground-based" observations, or convince me otherwise.

- Figure 1 caption—what does the blue color scale represent in Fig. 1a? I'd briefly mention it in the caption.

- Data availability: There was no data source given for the University of Canterbury ceilometer (CL51) data. Please include it as follows: https://doi.org/10.26179/5d91835e2ccc3 . In addition, no data sources were given for the radiometers or the all-sky cloud camera. The ARM data availability statement suffices for the ARM instruments, but if a DOI exists for these AAD instruments, they should be listed too.

- Perhaps a *little* more could be said about cloud phase in the last 2 paragraphs of Section 6 and how more or less supercooled liquid in the model relative to observations can impact your results. In general, the discussion of cloud microphysics in the manuscript is rather weak, and providing some speculative pathways for explanation would be very helpful.

**Line-specific Comments:**

**Line 108:** "was" should be "were"
**Line 110:** The two clauses should be joined by a conjunction, not a comma.
**Line 136:** I think you have to be careful when saying that supercooled clouds are typically not visible in the backscatter profile. Liquid-based supercooled clouds can exist at rather low altitudes (< 1-2 km AGL) over Macquarie Island, and in the absence of an underlying layer, will show a sharp gradient in attenuated backscatter consistent with a liquid cloud base identification. I would also mention that fog can frequently be observed in the backscatter profiles.
**Line 216:** Please give some reference to what type of CBH algorithm is used, since these are not trivial nor converged methods. I assume it is what is described in Section 5.3 of Kuma et al. (2021), and if so, I'd list the thresholds for attenuated volume backscattering coefficient that were employed here.
**Lines 270-271:** This statement is very confusing to me. I'm not sure what you mean by large spreads in the $LW_{cs}$ distributions, as the violin plot in Fig. 3d doesn't really show spread that is larger in model/satellite relative to ERA5. Also not sure what you mean by "paying more attention to the $LW_{cs}$ models than the $SW_{cs}$ models". Are you suggesting that $LW_{cs}$ biases are more important than $SW_{cs}$ biases, or appear to have a higher sensitivity? Please clear this up.
**Line 275:** Careful with this statement. "Too few, too bright" refers to compensating errors. All else being equal, "too bright" should *reflect* more surface SW radiation, so this component can't really be linked to an overestimation of absorbed surface SW.
**Lines 289-291:** Aside from the absolute values being ~ 2 W/m^2 off, the values you listed do seem consistent with Hinkelman and Marchand (2020)---SW bias of +8 for your study and +10 for their study, and LW bias of -12 for your study and -10 for their study. Understanding that 2 W/m^2 is not a small amount, I'd suggest rewording because "consistency" usually refers to how biases operate in sign, even if magnitudes are different.
**Line 299:** It's not necessary for this study, but perhaps the frequent soundings released at Macquarie Island can give you an idea of potential humidity and temperature biases in ERA5.

**Line 302:** It's not clear to me what you mean by three algorithms used for SW_cs biases. Are you including the ERA5 SW_cs calculations in this statement?

**Line 304:** Again, I wouldn't use "in-situ" here, as I think much of the community thinks of these as passive remote sensing instruments.

**Lines 301-305:** This paragraph seems a little out of place as you start discussing CREs. I expected this to be a transition to the next section, but that also wasn't clear because the paragraph ends on a "future guidance" type of statement. Suggest making the transition more clear or moving this paragraph to the next section.

**Line 316:** You haven't yet used the term "downward CRE" and it's not used anywhere else in the manuscript, so I'd avoid it here to avoid confusion.

**Line 323:** You say larger negative values of CRE_LW, but CRE_LW values are strictly positive using your convention.

**Line 355:** Fig. 6e does not show an overestimation in CF in spring as stated here, but rather the same mean CF for ACCESS-AM2 and observations.

**Line 360:** Is there a reason for suggesting cloud microphysics are a lesser control on the CRE compared to other properties? I would add some references to back this up if so. Droplet radius and size distributions are inherently linked to the cloud's optical thickness, after all.

**Line 382:** "results" should be singular "result"

**Line 385:** Suggest using "While much of that is…" instead of "While a lot of that is…"

**Line 473:** Saying "other" CFs is rather vague. I would provide a little more detail here and give the range where the model produced lower CF (between 0.2 and 0.6), or just say "lower" CFs instead.

**Line 484:** I think you mean under positive SW *bias* conditions. Also, "result" should be "results".

**Line 488:** Suggest making it clear that you are referring to a greater dependence on low-level CFO for LW biases *compared to* SW biases, if that is indeed what you mean.

**Line 489:** I wouldn't say "inappropriate" cloud representation. It's as appropriate as can be given scale separations, but must inevitably be parameterized. I'd suggest using "parameterized" instead of "inappropriate".

**Line 499:** Again, wouldn't use "in-situ"

**Line 523:** Should be either "These analyses suggest" or "This analysis suggests".

**References:**

Tansey, E., Marchand, R., Protat, A., Alexander, S. P., and Ding, S.: Southern Ocean Precipitation Characteristics Observed From CloudSat and Ground Instrumentation During the Macquarie Island Cloud & Radiation Experiment (MICRE): April 2016 to March 2017, Journal of Geophysical Research: Atmospheres, 127, e2021JD035 370, https://doi.org/10.1029/2021JD035370, 2022.

---

## Author Comment (AC1)

**Response to Reviewer 1:**

**Overall comments:**

*Pei et al. compare surface observations of downwelling shortwave and longwave radiation at Macquarie Island to the ACCESS-AM2 model and CERES product. They then analyze radiative biases as a function of cloud fraction and cloud occurrence, using the ALCF simulator (relying on hourly output from the model) and an all-sky cloud camera for the observational component. Overall, the authors clearly relate respective CRE biases to the identified shortcomings of both ACCESS-AM2 & CERES, in comparison to the MICRE observations. They establish a strong relationship between cloud fraction and SW/LW at the surface in both the model and observations. Interpretation of the role played by cloud occurrence (especially low clouds) is weaker, and I recommend that the authors provide more background information on the role of cloud microphysics (beyond cloud phase). This may include a brief survey of studies of Southern Ocean cloud properties from surface campaigns and in-situ aircraft measurements (e.g. during SOCRATES). I also recommend that the authors report the fraction of (and reasons for) missing/bad data for all observations (and whether you expect these to introduce any sampling bias), and more detail on the limitations of using ERA5 for clear-sky radiation estimates. I recommend that this paper be published after minor changes.*

We would like to thank Emily for your detailed comments. We have now provided more discussion related to cloud microphysics and reported the fraction of missing/bad data for all observations. Moreover, we have provided more details on the limitations of ERA5 radiation estimates. More information is outlined below in the comment-by-comment response.

**General comments:**

1) *In the introduction section second paragraph, more could be said about the sparseness of surface-based SO ceilometer observations (cloud occurrence frequency and cloud boundaries) and sources of uncertainty within the ALCF data. You state that this is a recently developed tool. Include a brief description of the ALCF here (with more detail in methods).*

We have added an acknowledgement of the description for sparseness of surface-based SO ceilometer, including the point that a lot of ceilometers have been installed in various locations worldwide, but not over the SO.

Line 39: 'Large networks of lidars and ceilometers have been installed globally, for instance, Cloudnet (Illingworth et al., 2007), E-PROFILE (Illingworth et al., 2019), and ARM (Campbell et al., 2002). Nevertheless, the surface-based ceilometer observations of cloud frequency of occurrence and cloud boundaries over the SO remain sparse (Kuma et al., 2020).

Source of ALCF uncertainty has been added in Section 2.6

Line 256: 'Several limitations exist within the ALCF that can cause uncertainties (Kuma et al., 2021). (1) The accuracy of the CL31 and CL51 ceilometers' calibration may be impacted by the absorption of water vapour at 910 nm, which can limit the precision of their comparison. However, it is improbable that the calculated cloud masks will be significantly influenced due to the high backscattering caused by clouds; (2) Precipitation and aerosol are not currently implemented in the simulator. The cloud detection algorithm typically identifies observed precipitation as "cloud", whereas the simulated profile does not show any backscattering in the area where precipitation is occurring. Upon reviewing the backscatter profiles, certain layers beneath stratocumulus clouds are identified as clouds, potentially consisting of drizzle, snow, fog, or aerosol. Nevertheless, the frequency of such occurrences is insufficient to significantly impact the statistics in a manner comparable to the model bias; (3) The ALCs also encounter several measurement limitations. Specifically, inadequate overlap, dead time, and after-pulse corrections often yield sub-optimal outcomes at close range. Semi-automated methods such as calculating the distribution of integrated attenuated volume backscattering coefficient by analyzing the height where maximum backscattering occurs.'

We also added further description of ALCF here.

Line 42: 'This is accomplished by extracting two-dimensional profiles (time x height) from the model data, using a modified COSP lidar simulator to perform radiative transfer calculations, calibrating and resampling the observed attenuated volume backscattering coefficient to a common resolution, and conducting similar cloud detection on both the simulated and observed attenuated volume backscattering coefficient (Kuma et al., 2021).'

2) *Regarding uncertainty, data issues and missing data fractions for all surface-based measurements: You provide a good description of this for the radiometer. For the MICRE ceilometer, can you provide a % of time that you do not have good data (missing a successful cloud base retrieval) because of fog, snow on the detector or other reasons for missing data? How does fog and attenuation affect your CFO profiles? (See also "general comment" #4 below.) In reference to the all-sky cloud camera, can you report the uncertainty and bad/missing data %?*

Times of missing/bad data from the ceilometer have been provided.

Line 160: 'During the selected period for conducting the radiation-cloud occurrence analysis in Section 5, which spanned from September 2017 to February 2018, approximately 6.7% of the ceilometer data were excluded due to poor quality.'

The effect of fog and attenuation on CFO profiles has been investigated by visually checking the ceilometer backscatter coefficient profiles over the whole period. We found that the precipitation and fog occur very rarely and will likely not influence the statistics.

Line 262: 'Precipitation and aerosol are not currently implemented in the simulator. The cloud detection algorithm typically identifies observed precipitation as "cloud", whereas the simulated profile does not show any backscattering in the area where precipitation is occurring. Upon reviewing the backscatter profiles, certain layers beneath stratocumulus clouds are identified as clouds, potentially consisting of drizzle, snow, fog, or aerosol. Nevertheless, the frequency of such occurrences is insufficient to significantly impact the statistics in a manner comparable to the model bias. Stanford et al. (2023) found ceilometer on Macquarie Island was obscured 2.5 % of the time because of fog.'

In reference to all-sky cloud camera, we exclude data coincident with missing radiometer measurements. There are no other specific processes to remove bad/missing data.

Line 173: 'The cloud camera dataset was organized to align with the available radiometer dataset, ensuring that the measurement of CF could be directly linked with radiation data.'

3) *In methods section 2.3.2, you state that you're able to contain clear-sky radiative biases in equation 4 by relying on ERA5, which ACCESS-AM2 & CERES also rely on. But this re-emerges later in the paper as a driver of uncertainty in the interpretation of results. Are you aware of any literature worth mentioning (whether early in the paper or in the discussion section) that provides more information on ERA5 radiative biases over the region? I scanned briefly and found one very recent article, for example:*
*Mallet, M. D., S. P. Alexander, A. Protat, and S. L. Fiddes, 2023: Reducing Southern Ocean shortwave radiation errors in the ERA5 reanalysis with machine learning and 25 years of surface observations. Artif. Intell. Earth Syst., https://doi.org/10.1175/AIES-D-22-0044.1*

We have added some discussion on ERA5 uncertainties. Also, we pointed out that we should pay attention to the radiation evaluation of model, satellite, and reanalysis products under clear-sky conditions.

Line 361: 'Wang et al. (2020) evaluated the cloud radiative effect of ERA5 using ship-based measurements in the SO during three summer seasons. Higher shortwave cloud radiative effect (+77 W m$^{-2}$) and lower longwave cloud radiative effect (-18 W m$^{-2}$) were detected in ERA5 in all-sky conditions, which are likely attributed to the higher occurrence of clouds over the Southern Ocean compared to what was modelled, and potentially resulting from the higher transmittance of clouds in the ERA5 (Wang et al., 2020). Regarding clear-sky conditions, no notable error was found in the ERA5 LW irradiance, while for SW, the observed values were 33 W m$^{-2}$ higher than those predicted by ERA5. More recently, Mallet et al. (2023) found large downwelling SW radiation biases (+54 W m$^{-2}$) in the ERA5 compared with 25 years summertime surface measurements collected from ship and ground station over the SO. By employing machine learning techniques, cloud cover and relative humidity exhibited a strong contribution to the SW radiation biases. Despite these few studies on ERA5 radiation biases, a limited amount of research has been dedicated to investigating this issue, particularly in relation to clear-sky conditions.'

4) *Figures 8 & 9 are of concern because of the very large CFO near the surface. Describe in detail how you got this profile and how you calculate cloud fractions at each height. Especially near the surface, >50% cloud occurrence in the lowest 50 meters seems improbable. If your approach was to define a minimum threshold on the ceilometer backscatter, you should discuss (and potentially rethink) this.*

We have added more details about how ALCF operates in Section 2.6. The cloud frequency of occurrence is calculated for each height level by counting the number of bins which have a positive cloud mask divided by the total number of columns in the time range. The cloud fraction is calculated by counting the number of columns which have at least one cloudy bin, divided by the total number of columns in the time range. To be detected, the cloud fraction of ceilometer and model outputted by ALCF is not the same as those observed by cloud camera, as ceilometer observes clouds vertically and cloud camera obverses a much larger region. For backscatter threshold for detecting clouds, we use 2 × 10$^{-6}$ m$^{-1}$ sr$^{-1}$ as this value was found to be a good compromise between false detection and misses in Southern Hemisphere (Kuma et al., 2021). Moreover, a number of standard deviations of noise at the given level is subtracted from the signal before the threshold is applied.

Line 240: 'For the model data, ALCF first extracts two-dimensional cloud liquid and ice content profiles at the survey area, then uses Subgrid Cloud Overlap Profile

Sampler (SCOPS) to generate 10 random subcolumns for each profile to detect clouds in the model (Chepfer et al., 2008). The default setting for generating cloud overlap is maximum-random overlap assumption, which assumes neighboring layers with non-zero CF are fully overlapped, while layers separated by zero CF are randomly overlapped. The same sampling rate (5 min) and vertical bins (50 m) were used in lidar simulator to make the model and observations comparable. The attenuated volume backscattering coefficient profiles are then simulated for 10 subcolumns based on the COSP lidar simulator. Subsequently, ALCF re-samples the observational profiles to increase the signal-to-noise ratio, subtracts the noise, calculates the lidar ratio, applies an absolute calibration, and uses a cloud detection algorithm to calculate cloud mask and CBH for both simulated and observational data. A threshold of $2 \times 10^{-6}$ m$^{-1}$ sr$^{-1}$ for backscattering coefficient is applied to identify cloud mask, as this value was found to be a good compromise between false detection and misses in Southern Hemisphere, where the data is less impacted by anthropogenic aerosol. This step is important to make the simulated and observed backscattering coefficient profiles comparable. Next, the statistical summary including CF, cloud frequency of occurrence (CFO) and attenuated volume backscattering coefficient histograms are derived. The CFO is calculated for each height level by counting the number of bins which have a positive cloud mask divided by the total number of columns in the time range. The total CF is calculated by counting the number of columns which have at least one cloudy bin, divided by the total number of columns in the time range. For the ceilometer data, ALCF applies the same operations as the model but starts from the denoised step. Plots of cloud occurrence representing the CBH and attenuated volume backscattering histogram are generated from the ALCF code. More information about this framework can be found in Kuma et al. (2020).'

5) *In the interpretation of results in section 5, I think you should discuss other possible sources of radiative biases here (e.g. differing cloud macro and microphysical properties). It is not sufficient to state only that the low-level CFO was less negative for negative SW biases, and provide no additional reasoning. If two opposing extremes of SW biases are both happening with underestimated low-level CFOs, additional cloud properties must play a role here.*
*Lines 502-505 start to address this. I think the discussion (and the paper overall) would be strengthened by referencing additional studies of cloud radiative effects as a function of macro and microphysics.*

We have provided more discussion on cloud microphysics in Section 5.

Line 531: 'Despite the large influence of cloud macrophysical characteristics such as CF and CFO on cloud radiative properties, it is essential to acknowledge the crucial role played by cloud microphysical properties such as cloud phase, cloud droplet number concentration, and cloud effective radius. This is particularly

important as two opposing extremes of SW biases are both observed with underestimated low-level CFOs. Vergara-Temprado et al. (2018) suggested the significance of incorporating the spatial and temporal variations of ice nucleating particle (INP) concentrations in cloud microphysics scheme. More realistic INP distributions and cloud microphysical properties are crucial to accurately simulate cloud phase, cloud reflectance and thus radiations (Tan and Storelvmo, 2016; Furtado and Field, 2017). Gettelman et al. (2020) compared cloud microphysics in a nudged global climate model (the Community Atmosphere Model, CAM) with aircraft observations (the Southern Ocean CLouds, Radiation, Aerosol, Transport Experimental Study, SOCRATES) collected over the SO. An improved simulation of SW CRE was shown by implementing a revised autoconversion scheme that reduces both liquid and ice water path but increases cloud fraction and effective radius, maintaining more supercooled liquid water. Nevertheless, the model still fell short of matching the droplet numbers observed in aircraft measurements, which suggests that higher concentrations of cloud condensation nuclei (CCN) and greater droplet numbers may be required to achieve better agreement (Gettelman et al., 2020). In light of these preceding studies, proper cloud macro and microphysical properties are necessary to correctly simulate the radiation balance in climate models.'

6) *You have two years' worth of seasonal cycles, and yet you do not discuss the differences in the two years. It would be interesting to see how much downwelling radiation, cloud fraction and vertical occurrence frequencies varied in each season during 2016-2017 as compared to 2017-2018.*

We have checked the differences in radiation and cloud fraction for the two years. Since we only possess CFO data for the summer of 2017-18, it is not possible to examine the seasonal variations in CFO.

[Figure]

Figure 1: Violin plot of SW and LW radiation in ACCESS, CERES, and Observation for 2016-17 and 2017-18

[Figure]

Figure 2: Same as Figure 1 but divided by seasons.

[Figure]

Figure 3: Violin plot of CF in ACCESS and Observation for 2016-17 and 2017-18

[Figure]

Figure 4: Same as Figure 3 but divided by seasons.

From Figure 1 we can see both SW and LW for the three datasets are similar, with observed SW in year 2 are slightly higher than in year1. For seasonal differences in Figure 2, SW and LW for all datasets basically look comparable over the two years, and there is an underestimation of observed SW in summer in year 1. For CF, the differences in two years are smaller (Figure 3,4).

Based on the results above, explaining these differences would require significant additional investigation. Therefore, we have decided to not include these yearly differences as part of the conclusions presented in this manuscript.

**7) Appendix B: What about diurnal errors in the CERES SYN product?**

We have checked the diurnal biases in CERES SYN product and have updated the plot in Appendix (Figure 5). We found large underestimated LW CRE in CERES at local night, which has been attributed to wrong cloud base height in previous studies.

[Figure]

Figure 5: Diurnal CREs biases in ACCESS-AM2 (red) and CERES (blue).

Line 398: 'For CERES, $CRE_{SW}$ biases are comparable to ACCESS-AM2, while the negative $CRE_{LW}$ biases are substantial at local night, which has been attributed to wrong cloud base height (Hinkelman and Marchand, 2020). The diurnal cycle highlights that in summer the ACCESS-AM2 model is able to more accurately capture the characteristics of LW radiation than it can for SW radiation, and most of the cloud radiative biases can be attributed to poor simulation of SW radiation. While in CERES, poor SW simulation during the day and LW simulation during the night both contribute to the total cloud radiative biases.'

Line 633: 'CERES $CRE_{SW}$ bias exhibits similarities to ACCESS-AM2, while it is larger during specific periods, such as 4 a.m. UTC, 19-21 p.m. UTC. Different to ACCESS-AM2, CERES $CRE_{LW}$ has notable negative biases at night local time (7 a.m. - 18 p.m. UTC), with biases ranging from -20 to -15 W m$^{-2}$. The significant underestimation of $CRE_{LW}$ in CERES, as highlighted in Hinkelman and Marchand (2020), is attributed to incorrect cloud base height during local nighttime periods.'

**Line-specific comments:**

1) **Line 50-52: You should also acknowledge here that discrepancies even exist between surface and satellite observations, due to limitations of satellite near-surface cloud retrievals.**

   We have added the discrepancies between surface and satellite observations.

   Line 60: 'However, discrepancies do exist between surface and satellite observations due to limitations of near-surface cloud retrievals of satellite.'

2) **I suggest connecting cloud feedbacks to the previous sentences; what cloud feedbacks are relevant to this study?**

   The poorly simulated cloud feedbacks caused by wrong cloud fraction and phase representation are related to our topic - radiation evaluation of the model. We have modified the sentence as follows.

   Line 64: 'Additionally, the poor representations of cloud feedbacks attributed to the reduction in low cloud coverage and water content lead to higher climate sensitivity in the Coupled Model Intercomparison Project phase 6 (CMIP6) compared to the previous version.'

3) **Line 62-63: Suggest rephrasing "remote atmospheric environment" to "the harsh atmospheric environment and lack of remote sites for measurements"**

   Sentence changed as suggested.

4) **Line 99-101: This explanation makes it difficult to follow what the time resolution is. 1-minute means and standard deviations? I suggest stating this up front.**

   Sentence been modified as follows.

   Line 114-116: 'The sensors have a time resolution of 1 minute, whose results were recorded as means and standard deviations for each of the 600 individual readings of output voltage at 1 minute interval, and logged on a Campbell Scientific CR3000 data logger.'

5) **Line 104: I suggest writing out "microvolts"**

   We will stick to convention for the unit of radiometer sensitivity $\mu V/(W\ m^2)$.

6) **Line 107: Please describe the calibration process in more detail. How did you get these coefficients? How do you determine sensitivities?**

We get these coefficients and sensitivities from the Kipp & Zonen radiometers certificate.

Sentence has been modified as follows.

Line 122: 'where R is the resistance ($\Omega$) and $\alpha$: $1.0295 \times 10^{-3}$, $\beta$: $2.391 \times 10^{-4}$, $\gamma$: $1.568 \times 10^{-7}$ are calibration coefficients from the Kipp & Zonen calibration certificate.'

7) *Line 119: Consider relating the 0.5°C to a % error in the measurements. Presumably this will have a very minimal effect on equation (1) since Tb is in Kelvin, although it is raised to the fourth power.*

We have calculated and changed 0.5 °C to 1 % error and have modified sentences as follows.

Line 137: 'Temperatures differences were within 1% between the two thermistors on average.'

8) *Line 123: "limited clipped points"*

Sentence changed as suggested.

9) *Line 127-129: In my opinion this should be stated earlier in the section.*

These sentences have been moved to the preceding paragraph. See Line 129-131.

10) *Line 133: Do you decrease 6-second sampling to 1-minute resolution to match up with other data? If so, what is your approach (such as using the median CBH, do you require a minimum detection threshold i.e. X columns out of 10 columns in the minute contain cloud base retrievals?)*

We sub-sampled ceilometer data to 5-minute time resolution and 50-meter vertical resolution by averaging columns and bins through ALCF. We didn't require a minimum detection threshold. This information has been added as follows.

Line 149: 'The ceilometer observations were sub-sampled to 5-minute time resolution and 50-meter vertical resolution by averaging multiple columns and bins through ALCF (Kuma et al., 2020).'

11) *Line 190: Explain why September-February were chosen to look at hourly simulations.*

September-February 2017-18 was chosen to look at hourly simulations in aims to matching three other campaigns described in McFarquhar et al. (2021) besides MICRE. Information been added as follows.

Line 210: 'Model output has been saved as daily means from April 2016 to March 2018, and limited hourly instantaneous output from September 2017 to February 2018 to coincide with three other campaigns described in McFarquhar et al. (2021) besides MICRE.'

12) **Line 212-214: Provide some comment on the input to the ALCF simulator.**

The input to the ALCF simulator has been provided as follows.

Line 236: 'It conducts the required steps to model the ALC attenuated volume backscattering coefficient by extracting cloud liquid and ice mixing ratios, cloud fraction, and thermodynamic data from the model.'

13) **Line 217: misspelled ceilometer**

Sentence changed as suggested.

14) **Line 218-219: What is the spacing of height bins? What is a column? E.g. a 1-minute vertical profile of 100-meter vertical bins extending up to 15km?**

We provided the definitions of bin and column in Section 2.2.2 as follows.

Line 150: 'Columns and bins here are time and vertical intervals of the backscatter profile.'

By referring to the previous sentence, here the spacings of bins and columns are respectively 50m and 5min.

15) **Line 233: "Generally consistent" - I suggest making this statement quantitative: ACCESS, CERES match observations to within X,Y in W/m^2, "as seen in table 1". You start to do this at line 250, but a comment on general consistency can be made earlier to prepare the reader for what constitutes a significant disagreement.**

Sentences been modified as follows.

Line 280: 'The surface SW radiation fluxes simulated by ACCESS-AM2 model and CERES align with observations regarding the $R^2$ values of 0.79 and 0.93 respectively (Figure 2a)'

**16) Line 237: Curiously, this underestimation seems more pronounced in the second winter (JJA 2017) when I look at Figure 2. Is this the case? If so, perhaps comment on this in the text?**

From Figure 2 in this response, we can see the LW of CERES product in the second winter doesn't exhibit a greater degree of underestimation compared to first year's winter.

**17) Line 254-255 (and re: Figure 3 in general): Perhaps add a sentence here or in the figure caption explaining the violin plot spread at various radiative fluxes (vertical axis values), and what the bolded segment is meant to show on the middle line.**

We have added the explanation for the violin plot in the Figure 3 caption: 'The white dot on the middle represents the median, the thick gray bar represents the interquartile range, and the thin gray line represents the rest of the distribution. The width of the violin plot represents the distribution of radiation value.'

**18) Line 256: I suggest a rewrite, "reaching -4 W/m^2 in Autumn, with smaller differences in all other seasons."**

Sentence changed as suggested.

**19) Line 275-276: In more recent generations this "too few, too bright" issue has been modified -- perhaps talk about findings from Schuddeboom and McDonald (2021). Are the cloud fraction differences discussed in agreement with this analysis?**
**Schuddeboom, A. J., & McDonald, A. J. (2021). The Southern Ocean radiative bias, cloud compensating errors, and equilibrium climate sensitivity in CMIP6 models. Journal of Geophysical Research: Atmospheres, 126, e2021JD035310. https://doi.org/10.1029/2021JD035310**

We have added the discussion of Schuddeboom and McDonald (2021)'s work as follows. Our overall overestimated CF in the model agrees with this analysis.

Line 327: '"Too few and too bright" low-level clouds were identified as the cause of this SW bias in CMIP5 models (Nam et al., 2012; Wall et al., 2017). Nevertheless, more recently, Schuddeboom and McDonald (2021) discovered the exact contrasting result in the CMIP6 simulations, which demonstrates the importance of prioritizing the low-level cloud simulation to enhance the SW radiative balance over the SO.'

**20) Line 289 You write that the biases found here (SW +8.0 ± 18.0Wm−2 , LW bias of -12.1 ± 12.2Wm−2 ) are "NOT consistent" with the previous study by Hinkelman**

*and Marchand (2020) (SW +10Wm−2 , LW -10Wm−2 ). How so? The means would appear to be nearly the same, and well within the listed uncertainty? In general, I don't understand the ± values given here, as they don't seem to match table #1.*

These ~2 W m$^{-2}$ differences in SW and LW are possibly attributed to different temporal resolutions of the CERES product used (daily output used in this manuscript and hourly output used in Hinkelman and Marchand (2020)) and different interpolation method to assign data to Macquarie Island (we linearly interpolated data to Macquarie Island while Hinkelman and Marchand (2020) chose the nearest grid that contains Macquarie Island). And other effects such as data gaps, calibration offsets, and local shadowing effects may also influence the biases. Sentences been modified as follows.

Line 344: 'These differences in SW & LW biases are possibly attributed to different temporal resolution of CERES SYN product (hourly output used in Hinkelman and Marchand (2020) and daily output used in this study) and different interpolation methods to collocate data to Macquarie Island (Hinkelman and Marchand (2020) chose the nearest grid that contains Macquarie Island while this study linearly interpolated data to Macquarie Island). Other factors such as data gaps, sampling uncertainty, calibration offsets, different pyranometers, and local shadowing effects may also contribute to the biases difference.'

The ± values here indicate the standard deviation of biases. While in Table 1, unbolded brackets indicate the standard deviation of mean value and bolded brackets indicate the standard error of mean difference, which reflect if the biases can be considered as significant at a certain confidence interval.

21) *Line 292: Isn't the ARM pyranometer from Hinkelman & Marchand [2020] the one used for the comparisons in Figs. A1 & A2? One would think calibration offsets and local shadowing effects have only minor impacts since the comparisons in the appendix are in such good agreement. Also earlier you said only 9 days have been removed from your timeseries. So are data gaps in the ARM pyranometer more frequent?*

The potential reasons for the differences in biases have been suggested in comment 20). The data gaps for pyranometer (SW) of AAD and ARM are similar (~23 months). But for the pyrgeometer (LW), ARM's data (19 months) gaps are more frequent than AAD's (23 months).

22) *Line 296. You write "Excellent alignment of SWcs radiation between the satellite and reanalysis is expected given the CERES product uses ERA5 to inform its radiative transfer algorithm." Perhaps start a new paragraph as this represents a change in topic.*

We have made this a new paragraph.

23) *Line 312-313: I would argue from downwelling SW in Fig. 3d that the winter and spring clear-sky LW are somewhat lower, exhibiting a weak but discernible seasonal cycle. Suggest removing "clear-sky" or changing to "similar to the all-sky and, to a lesser degree, clear sky"*

We have removed 'clear-sky'.

24) *Line 315: "Figure 5 shows"*

Sentence changed as suggested.

25) *Line 319: "During winter, when total CRE is at the lowest value" – JJA is the season where the CRE is highest (the most positive value). And if you are talking about magnitudes only, it is lower in MAM than in JJA. I'm not sure what you're referring to here.*

We refer that the winter CRE is the most positive. We have changed 'lowest' to 'most positive'.

26) *Line 325: Suggested that you change "climate models" to "ACCESS-AM2"*

Sentence changed as suggested.

27) *Line 329: "which peaks"*

Sentence changed as suggested.

28) *Caption of Figure 6 & line 341: Use the plural, "observations"*

Sentence changed as suggested.

29) *Line 342: Draw the reader's attention to specifically Fig. 6a to demonstrate the integrated effect discussed in this sentence.*

We have added '(Figure 6a)' at the end of this sentence.

30) *Line 343: Only Figure 6e is for spring.*

We have specified Figure 6e in this sentence.

31) *Line 350: "mid-level CF": You might include the thresholds here (above 4 km / below 10 km or something).*

Here we meant 'intermediate' CF, rather than 'mid-level' CF. We have corrected the word.

**32) Line 357: "particularly"**

Sentence changed as suggested.

**33) Line 360: If you state that microphysics plays a lesser role in cloud radiative effect than the macrophysics described in the previous sentence, you should back this up with references. More references in general in lines 358-363 would be good.**

We have changed the statement that 'cloud microphysics plays a lesser role' and back up with more references. Sentences been modified as follows.

Line 430: 'Nevertheless, the overall overestimated CF and positive surface SW biases in the model indicate that the CF alone does not control the cloud radiative effect, but also properties such as cloud phase, cloud base height, and cloud geometrical or optical thickness are likely to play a significant role (Viudez-Mora et al., 2015; Cesana and Storelvmo, 2017; Fiddes et al., 2022). In addition, cloud microphysics such as ice crystal shape and size distribution and direct and indirect effect of aerosols could also have an effect on radiation biases (Bohren and Huffman, 2008; Kuma et al., 2020). Our results here are in agreement with the work done by Schuddeboom and Mcdonald (2021), which found overestimated low-level CF and reduced reflectivity of low-level cloud over the SO in CMIP6 models, highlighting the significance of correctly representing low-level clouds to simulating radiative balance over the SO.'

**34) Line 369: "thresholds"**

Sentence changed as suggested.

**35) Line 374: Period after "LW bias" should be a comma**

Sentence changed as suggested.

**36) Line 382: "This result"**

Sentence changed as suggested.

**37) Line 388-389: Over what number of hours are you calculating cloud fraction? How are you averaging and comparing these two different datasets with different resolutions?**

We calculated total CF from September 2017 to February 2018. We note that the CF here is not the mean CF of each hour over the period but is the percentage that there are clouds detected above the ceilometer through the period.

The ALCF makes two different datasets comparable by subsampling observation profiles to 5-min and 50-meter resolution and using the same vertical bins to detect clouds in the model. More descriptions have been added in Section 2.6.

**38) Line 389: "Figures"**

Sentence changed as suggested.

**39) Figure 8: I recommend adding shading around the vertical cloud fraction profiles indicating e.g. standard error on the mean CF at each height bin.**

The cloud frequency of occurrence profiles here are not averaged CF at each height but show the percentage of cloud occurrence at each height over a period. Thus, we are not able to get the standard deviation or standard error of CFO. For each time columns, the ALCF will simulate 10 random subcolumns. We took the averaged value of CFO and total CF of 10 subcolumns. The standard deviation of 10 subcolumns is checked, which is too small (0.2%) to be evident on plots. So, we decided to not show it in Figure 8 and 9.

**40) Line 395: "overleap" – you probably mean overlap**

Word corrected.

**41) Line 404: "summertime only" – The period is spring and summertime.**

Sentences has been modified as follows.

Line 479: 'The overall CF for this period (spring and summer) observed by the ceilometer was 94 %, which the model underestimated by 6 %.'

**42) Line 404: I suggest rephrasing to "which the model underestimated by 6%."**

Sentence changed as suggested.

**43) Line 405-406: Look for references on how often multi-layer clouds occur. This can give you some estimate of how frequently the ceilometer suffers this drawback of high cloud obscurity.**

References for how often multi-layer clouds occurs over the SO have been added as follows,

Line 484: 'Multilayer cloud occurrence of 19.5 % was obtained by Protat et al. (2017) within a span of 10 days between latitudes of 43°S to 48°S. Klekociuk et al. (2020) found a 26 % occurrence of multilayer cloud during a two-month campaign from latitudes 44.7°S to 67°S. By examining these previous observations, we can have an approximation of the frequency at which the ceilometer experiences the limitation of high cloud obscurity over the SO.'

44) *Line 408: I'm not sure I follow this. Why would ACCESS-AM2 output passed through ALCF suffer the same backscatter attenuation limitation?*

Referred from Kuma et al. (2021) Section 4: 'The simulation is implemented by applying the lidar equation on model levels. Scattering and absorption by cloud particles and air molecules are calculated using the Mie and Rayleigh theory.' Thus we said that the model also suffers the backscatter attenuation limitation.

45) *Line 422-425: Perhaps include a range or estimate of the uncertainty on the 4% higher total CF. In any case, especially considering the missing low-cloud occurrence < 1km, this result is surprising. Additionally, I cannot tell by looking at 9a that the red line yields 4% higher CFO than the black line, all vertical levels considered... But this could just be an effect of the vertical axis log scale.*

As we responded in comment 37), the total CF here is not the mean CF throughout the period but the percentage that there are any clouds detected above the ceilometer and model over the period. Thus, we are not able to give the uncertainty of the total CF outputted by ALCF. In vertical levels, the profile only tells the cloud frequency of occurrence at each level but not the CF, so the 4% higher total CF in the model can't be observed in the vertical profile.

46) *Line 425: I suggest changing "Figure 8" to "as seen in Figure 8 and replotted in Figure 9 in lighter colors"*

Sentence changed as suggested.

47) *Line 428: "model's"*

Sentence changed as suggested.

48) *Line 433: I suggest changing "simulates" to "overestimates"*

Sentence changed as suggested.

49) *Line 439: Awkward and needs rewording. I suggest changing to "Below 500m, the modeled cloud occurrence is lower than the average,"*

Sentence changed as suggested.

**50) Line 444: "can be in the model"? Maybe reword this.**

We changed it to 'can be simulated by the model'.

**51) Line 446: "cloud phase" – Cloud microphysics in general are likely contributing to SW radiative biases, including cloud droplet size and number concentration, cloud optical depth.**

We removed this sentence and provided more discussion about cloud microphysics starting from Line 525.

**52) Line 470-471: Given the dependence on ERA5, can you speculate on the source of this problem?**

We have provided the speculation for source of ERA5 biases as follows.

Line 564: 'We speculate that temperature and humidity representation play an important role in causing the LWcs bias in CERES, and suggest that further research should be conducted to evaluate clear-sky radiation properties in CERES and ERA5.'

**53) Line 472: "observations"**

Sentence changed as suggested.

**54) Line 472-473: Elaborate more on these compensating errors as a separate sentence.**

We have elaborated more as follows.

Line 567: 'However, this is caused by an underestimated frequency of CF between 0.2 and 0.6 and an overestimated frequency of CF above 0.6.'

**55) Line 473: Remove the comma after "prior"**

Sentence changed as suggested.

**56) Line 489: "inappropriate" is a funny word choice; I suggest inadequate or erroneous.**

Word changed to 'erroneous' as suggested.

**57) Line 490-491: If data are missing during these periods, what makes you say there were outliers in the observations?**

Here, the outliers meant the missing data. Sentence been modified as follows.

Line 585: 'For example, observed missing data on specific days have been ignored.'

**58) Line 491-492: "which will include inaccuracies" – Such as?**

We have added the inaccuracies as follows.

Line 585: 'The simulated clear-sky radiations are based on the modelled ERA5 product, which will include inaccuracies on clear-sky radiation estimation due to non-optimal temperature and humidity profiles.'

**59) Line 508: "by further**

Sentence changed as suggested.

---

## Author Comment (AC2)

**Response to Reviewer 2:**

**Overall comments:**

*This manuscript evaluated cloud radiative biases in the ACCESS-AM2 model and the CERES satellite product against ground-based observations during the MICRE campaign at Macquarie Island. The study used a suite of instrumentation to establish a positive bias in surface shortwave radiation in both the model and CERES, which is consistent with numerous past radiative flux modeling studies over the Southern Ocean. The authors also employed a new lidar simulator to constrain model output with observational instrument specifications, therefore allowing an evaluation of cloud attribution to the radiative biases. They found that excessive absorbed surface shortwave radiation is associated with too low of a cloud fraction. Overall, the manuscript is well-written, descriptive where needed and brief where appropriate, and the experiment is well-designed. The largest concern is the lack of discussion of the potential role that cloud microphysics have in the discussed biases. While I understand this is not a microphysics study, there are long-standing microphysical biases in GCMs that are suggested to be relevant to simulating Southern Ocean clouds (e.g., prevalence and maintenance of supercooled liquid) that could be discussed in more detail as potential caveats. Otherwise, I have provided some general comments and line-specific comments below, none of which should require too much extra effort to delay publication. I suggest this manuscript be accepted for publication after these minor revisions are addressed.*

We thank Reviewer 2 for taking the time to comment on our manuscript. We have provided more discussions about the potential role of cloud microphysics have in the radiation biases. More responses for comments are as below.

**General comments:**

1) *In the second paragraph of the Introduction when you introduce the ALCF, I think some expansion is needed for unfamiliar readers of how this framework operates. Typically, simulators are thought of as being applied to model data, not on observational data, so this can be confusing and deserves a little more explanation. Based on your description in Section 2.6, it seems ALCF is used on the observational ceilometer dataset as more of a means of calibration rather than "simulation".*

Indeed, ALCF is not used as a simulator with observational data. It performs calibration, resampling, and cloud detection, in a way which is corresponding the

processing of model data. We have added more description of ALCF in the second paragraph of the Introduction and Section 2.6.

Line 43: This is accomplished by extracting two-dimensional profiles (time x height) from the model data, using a modified COSP lidar simulator to perform radiative transfer calculations, calibrating and resampling the observed attenuated volume backscattering coefficient to a common resolution, and conducting similar cloud detection on both the simulated and observed attenuated volume backscattering coefficient (Kuma et al., 2021).

Line 240: For the model data, ALCF first extracts two-dimensional cloud liquid and ice content profiles at the survey area, then uses Subgrid Cloud Overlap Profile Sampler (SCOPS) to generate 10 random subcolumns for each profile to detect clouds in the model (Chepfer et al., 2008). The default setting for generating cloud overlap is maximum-random overlap assumption, which assumes neighboring layers with non-zero CF are fully overlapped, while layers separated by zero CF are randomly overlapped. The same sampling rate (5 min) and vertical bins (50 m) were used in lidar simulator to make the model and observations comparable. The attenuated volume backscattering coefficient profiles are then simulated for 10 subcolumns based on the COSP lidar simulator. Subsequently, ALCF re-samples the observational profiles to increase the signal-to-noise ratio, subtracts the noise, calculates the lidar ratio, applies an absolute calibration, and uses a cloud detection algorithm to calculate cloud mask and CBH for both simulated and observational data. A threshold of $2 \times 10^{-6}$ $m^{-1}$ $sr^{-1}$ for backscattering coefficient is applied to identify cloud mask, as this value was found to be a good compromise between false detection and misses in Southern Hemisphere, where the data is less impacted by anthropogenic aerosol. This step is important to make the simulated and observed backscattering coefficient profiles comparable.

Line 258: Several limitations exist within the ALCF that can cause uncertainties (Kuma et al., 2021). Firstly, the accuracy of the CL31 and CL51 ceilometers' calibration may be impacted by the absorption of water vapour at 910 nm, which can limit the precision of their comparison. However, it is improbable that the calculated cloud masks will be significantly influenced due to the high backscattering caused by clouds. Secondly, precipitation and aerosol are not currently implemented in the simulator. The cloud detection algorithm typically identifies observed precipitation as "cloud", whereas the simulated profile does not show any backscattering in the area where precipitation is occurring. Upon reviewing the backscatter profiles, certain layers beneath stratocumulus clouds are identified as clouds, potentially consisting of drizzle, snow, fog, or aerosol. Nevertheless, the frequency of such occurrences is insufficient to significantly impact the statistics in a manner comparable to the model bias. Finally, the ALCs also encounter several measurement limitations. Specifically, inadequate overlap, dead time, and after-pulse corrections often yield sub-optimal outcomes in the

close range. Semi-automated methods include calculating the distribution of integrated attenuated volume backscattering coefficient by analyzing the height where maximum backscattering occurs.

2) *In the fourth paragraph of the introduction, I think something should be said about satellite limitations in observing low-level clouds over the SO, which is a very strong motivation for evaluating ground-based observations. As one example, Tansey et al. (2022; https://doi.org/10.1029/2021JD035370) looked at surface precipitation measurements during MICRE in comparison with CloudSat, with a few notable differences based on satellite instrument sensitivities and algorithm structure.*

We have added statements for satellite limitations as follows.

Line 73: 'Tansey at al. (2022) examined surface precipitation measurements during MICRE and compared them with data from CloudSat, revealing several notable differences attributable to satellite instrument sensitivities and algorithm structure. This indicates the limitations of satellite in observing low-level clouds over the SO, which serves as a strong motivation for utilizing ground-based observations to calibrate satellite products.'

3) *You discuss how observed cloud fraction is defined by the all-sky cloud camera (Section 2.2.3) and then state it is used to evaluate CF biases on line 337 in Section 4, but it's not entirely clear how CF is being defined by ACCESS-AM2. Is it the prognostic value? Is it the CF computed by ALCF? It would be good to mention this at the beginning of Section 4.*

We have provided how CF is defined by ACCESS-AM2 in Section 2.4. It is the prognostic value from PC2 cloud fraction scheme.

Line 221: 'Of interest to this study, the ACCESS-AM2 model uses the Suite of Community RAdiative Transfer codes based on Edwards and Slingo (SOCRATES) (Edward and Slingo, 1996) and Wilson et al. (2008)'s prognostic CF and condensate cloud scheme, which includes large-scale as well as convective clouds. For comparison with the observational data, radiation and prognostic CF in the model was linearly interpolated to the point nearest to Macquarie Island (54.5°S, 158.9°E).'

4) *Section 2.1 and other parts of manuscript—I wouldn't call this in-situ observations, since many of what is included are more commonly thought of as passive remote sensing instruments. Perhaps change to "ground-based" observations, or convince me otherwise.*

We have changed all 'in-situ' to 'ground-based'.

5) *Figure 1 caption–what does the blue color scale represent in Fig. 1a? I'd briefly mention it in the caption.*

 We have mentioned in the Figure 1 caption: 'The blue color scale represents the bathymetry of oceans.'

6) *Data availability: There was no data source given for the University of Canterbury ceilometer (CL51) data. Please include it as follows: https://doi.org/10.26179/5d91835e2ccc3 . In addition, no data sources were given for the radiometers or the all-sky cloud camera. The ARM data availability statement suffices for the ARM instruments, but if a DOI exists for these AAD instruments, they should be listed too.*

We have added the data source for UC ceilometer in the *Data availability*. Data sources for AAD radiometers and cloud camera will soon be published on Australian Antarctic Data Centre (AADC).

Line 609: 'The AAD radiometer and all-sky cloud camera data will shortly be available from the Australian Antarctic Data Centre (AADC). The University of Canterbury's Vaisala CL51 ceilometer data are available at AADC (https://doi.org/10.26179/5d91835e2ccc3).'

7) *Perhaps a \*little\* more could be said about cloud phase in the last 2 paragraphs of Section 6 and how more or less supercooled liquid in the model relative to observations can impact your results. In general, the discussion of cloud microphysics in the manuscript is rather weak, and providing some speculative pathways for explanation would be very helpful.*

We have emphasized SLW's role in impacting the radiation biases in Section 6.

Line 600: 'We emphasize that the correct representation of supercooled liquid water over the SO is important for modelling the radiation in the region, as inadequate supercooled liquid water content will cause less reflectivity of clouds and result in positive downwelling surface SW biases (Luo et al., 2016; Vergara-Temprado et al., 2018; Gettelma et al., 2020).'

Moreover, we have provided more discussion of cloud microphysics in the last paragraph of Section 5, which starts from Line 531.

**Line-specific Comments:**

1) *Line 108: "was" should be "were"*

Word has been changed.

**2) Line 110: The two clauses should be joined by a conjunction, not a comma.**

Sentence has been modified.

**3) Line 136: I think you have to be careful when saying that supercooled clouds are typically not visible in the backscatter profile. Liquid-based supercooled clouds can exist at rather low altitudes (< 1-2 km AGL) over Macquarie Island, and in the absence of an underlying layer, will show a sharp gradient in attenuated backscatter consistent with a liquid cloud base identification. I would also mention that fog can frequently be observed in the backscatter profiles.**

We have deleted the 'supercooled cloud layers' in this sentence and added that fog can also be observed.

Line 151: 'Information on CBH, precipitation, and infrequently boundary layer height can be obtained from the backscatter profile using detection algorithms. Fog can be observed in the backscatter profiles as well.'

**4) Line 216: Please give some reference to what type of CBH algorithm is used, since these are not trivial nor converged methods. I assume it is what is described in Section 5.3 of Kuma et al. (2021), and if so, I'd list the thresholds for attenuated volume backscattering coefficient that were employed here.**

We have provided the information of thresholds for attenuated volume backscattering coefficient in Section 2.6. The threshold we used is $2 \times 10^{-6}\,\mathrm{m}^{-1}\,\mathrm{sr}^{-1}$, which was found to be suitable for the Southern Hemisphere.

Line 248: 'A threshold of $2 \times 10^{-6}\,\mathrm{m}^{-1}\,\mathrm{sr}^{-1}$ for backscattering coefficient is applied to identify cloud mask  after removing 5 standard deviations of range-scaled noise, as this value was found to be a good compromise between false detection and misses in Southern Hemisphere, where the data is less impacted by anthropogenic aerosol.'

**5) Lines 270-271: This statement is very confusing to me. I'm not sure what you mean by large spreads in the LW_cs distributions, as the violin plot in Fig. 3d doesn't really show spread that is larger in model/satellite relative to ERA5. Also not sure what you mean by "paying more attention to the LW_cs models than the SW_cs models". Are you suggesting that LW_cs biases are more important than SW_cs biases, or appear to have a higher sensitivity? Please clear this up.**

Here we tried to emphasize the larger biases in LWcs and that more work is needed in this space. We have modified the sentences as follows.

Line 318: 'The significant differences of LWcs in model and satellite compared to ERA5 highlight the need for more validation on development of especially the LWcs models. The SWcs models show smaller and insignificant biases, indicating less uncertainty.'

6) *Line 275: Careful with this statement. "Too few, too bright" refers to compensating errors. All else being equal, "too bright" should reflect more surface SW radiation, so this component can't really be linked to an overestimation of absorbed surface SW.*

Yes, too bright clouds will result in underestimated surface SW radiation. We have revised this statement as follows.

Line 327: '"Too few and too bright" low-level clouds were identified as the cause of this SW bias in CMIP5 models (Nam et al., 2012; Wall et al., 2017). Nevertheless, more recently, Schuddeboom and McDonald (2021) discovered the exact contrasting result in the CMIP6 simulations, which demonstrates the importance of prioritizing the low-level cloud simulation to enhance the SW radiative balance over the SO.'

7) *Lines 289-291: Aside from the absolute values being ~ 2 W/m^2 off, the values you listed do seem consistent with Hinkelman and Marchand (2020)---SW bias of +8 for your study and +10 for their study, and LW bias of -12 for your study and -10 for their study. Understanding that 2 W/m^2 is not a small amount, I'd suggest rewording because "consistency" usually refers to how biases operate in sign, even if magnitudes are different.*

We reworded the 'not consistent' to 'not equal to'. Moreover, we have added more speculation for this difference between two study.

Line 344: These differences in SW & LW biases are possibly attributed to different temporal resolution of the CERES SYN product (hourly output used in Hinkelman and Marchand (2020) and daily output used in this study) and different interpolation methods to collocate data to Macquarie Island (Hinkelman and Marchand (2020) chose the nearest grid that contains Macquarie Island while this study linearly interpolated data to Macquarie Island). Other factors such as data gaps, sampling uncertainty, calibration offsets, different pyranometers, and local shadowing effects may also contribute to the biases difference.

8) *Line 299: It's not necessary for this study, but perhaps the frequent soundings released at Macquarie Island can give you an idea of potential humidity and temperature biases in ERA5.*

The soundings data at Macquarie Island has been assimilated into the ERA5 product, thus we expect that the ERA5 data performs satisfactorily compared to the sounding observations at Macquarie Island.

9) **Line 302: It's not clear to me what you mean by three algorithms used for SW_cs biases. Are you including the ERA5 SW_cs calculations in this statement?**

Here, we meant ACCESS-AM2 and CERES used similar algorithms to ERA5. We have corrected the statement.

Line 359: 'Here we have shown that while the SWcs biases from ACCESS-AM2 and CERES (using similar meteorology driven by ERA5 and using the same method of calculating the clear-sky fluxes) are very similar, the same cannot be said for the LWcs.'

10) **Line 304: Again, I wouldn't use "in-situ" here, as I think much of the community thinks of these as passive remote sensing instruments.**

We have changed 'in-situ' to 'ground-based'.

11) **Lines 301-305: This paragraph seems a little out of place as you start discussing CREs. I expected this to be a transition to the next section, but that also wasn't clear because the paragraph ends on a "future guidance" type of statement. Suggest making the transition more clear or moving this paragraph to the next section.**

We added more discussions of ERA5 radiation biases.

Line 356: Understanding the biases in the respective SW and LW clear-sky biases is an important but often neglected component of understanding the CREs. Here we have shown that while the SWcs biases from ACCESS-AM2 and CERES (using similar meteorology driven by ERA5 and using the same method of calculating the clear-sky fluxes) are very similar, the same cannot be said for the LWcs. These differences, and how they affect the CRE, require further study. Wang et al. (2020) evaluated the cloud radiative effect of ERA5 using ship-based measurements in the SO during three summer seasons. Higher shortwave cloud radiative effect (+77 W m$^{-2}$) and lower longwave cloud radiative effect (-18 W m$^{-2}$) were detected in ERA5 in all-sky conditions, which are likely attributed to the higher occurrence of clouds over the Southern Ocean compared to what was modelled, and potentially resulting from the higher transmittance of clouds in the ERA5 (Wang et al., 2020). Regarding clear-sky conditions, no notable error was found in the ERA5 LW irradiance, while for SW, the observed values were 33 W m$^{-2}$ higher than those predicted by ERA5. More recently, Mallet et al. (2023) found large downwelling SW radiation biases (+54 W m$^{-2}$) in the ERA5 compared with 25 years summertime surface measurements collected from ship and ground station over the SO. By

employing machine learning techniques, cloud cover and relative humidity exhibited a strong contribution to the SW radiation biases. Despite these few studies on ERA5 radiation biases, a limited amount of research has been dedicated to investigating this issue, particularly in relation to clear-sky conditions. We suggest the importance of using ground-based observations of clear-sky radiation to evaluate the model and satellite, as well as validating the reanalysis product.

And we added a paragraph for transition as follows.

Line 374: 'After investigating the SW and LW radiation biases of ACCESS-AM2 and CERES in both all-sky and clear-sky conditions, we next assess their capability to reproduce cloud radiative effect.'

12) **Line 316: You haven't yet used the term "downward CRE" and it's not used anywhere else in the manuscript, so I'd avoid it here to avoid confusion.**

We have removed 'downward'.

13) **Line 323: You say larger negative values of CRE_LW, but CRE_LW values are strictly positive using your convention.**

Here, we meant the larger negative biases. We have changed 'values' to 'biases'.

14) **Line 355: Fig. 6e does not show an overestimation in CF in spring as stated here, but rather the same mean CF for ACCESS-AM2 and observations.**

Yes, we have deleted 'spring'.

15) **Line 360: Is there a reason for suggesting cloud microphysics are a lesser control on the CRE compared to other properties? I would add some references to back this up if so. Droplet radius and size distributions are inherently linked to the cloud's optical thickness, after all.**

Cloud microphysics also plays an important role in affecting radiation biases. We have modified the sentence and provided more references as follows.

Line 430: 'Nevertheless, the overall overestimated CF and positive surface SW biases in the model indicate that the CF alone does not control the cloud radiative effect, but also properties such as cloud phase, cloud base height, and cloud geometrical or optical thickness are likely to play a significant role (Viudez-Mora et al., 2015; Cesana and Storelvmo, 2017; Fiddes et al., 2022). In addition, cloud microphysics such as ice crystal habit and size distribution and direct and indirect effect of aerosols could also have an effect on radiation biases (Bohren and Huffman, 2008; Kuma et al., 2020). Our results here are in agreement with the work done by Schuddeboom and McDonald (2021), which found overestimated

low-level CF and reduced reflectivity of low-level cloud over the SO in CMIP6 models, highlighting the significance of correctly representing low-level clouds to simulating radiative balance over the SO.'

16) **Line 382: "results" should be singular "result"**

Word has been changed.

17) **Line 385: Suggest using "While much of that is..." instead of "While a lot of that is..."**

Sentence has been changed as suggested.

18) **Line 473: Saying "other" CFs is rather vague. I would provide a little more detail here and give the range where the model produced lower CF (between 0.2 and 0.6), or just say "lower" CFs instead.**

We have specified more details about 'other' CFs.

Line 567: 'However, this is caused by an underestimated frequency of CF between 0.2 and 0.6 and an overestimated frequency of CF above 0.6.'

19) **Line 484: I think you mean under positive SW *bias* conditions. Also, "result" should be "results".**

Words have been changed as suggested.

20) **Line 488: Suggest making it clear that you are referring to a greater dependence on low-level CFO for LW biases *compared to* SW biases, if that is indeed what you mean.**

Yes, we have modified the sentence as follows.

Line 582: 'We suggest that the larger differences in the modelled low-level cloud occurrences between the LW conditions demonstrates the greater dependence on low-level CFO of the LW biases than SW biases.'

21) **Line 489: I wouldn't say "inappropriate" cloud representation. It's as appropriate as can be given scale separations, but must inevitably be parameterized. I'd suggest using "parameterized" instead of "inappropriate".**

We reworded 'inappropriate' to 'erroneous' as suggested by Reviewer 1, as we want to stress the incorrect cloud parameterization in the model.

22) **Line 499: Again, wouldn't use "in-situ"**

We have changed 'in-situ' to 'ground-based'.

**23) Line 523: Should be either "These analyses suggest" or "This analysis suggests".**

Sentence has been changed as suggested.

---

## Referee Report (RR1)

Thanks for addressing my comments in detail with your revisions. Regarding my previous comment and request for clarification (pasted below):

*"Figures 8 & 9 are of concern because of the very large CFO near the surface.*

*Describe in detail how you got this profile and how you calculate cloud fractions*

*at each height. Especially near the surface, >50% cloud occurrence in the lowest*

*50 meters seems improbable. If your approach was to define a minimum*

*threshold on the ceilometer backscatter, you should discuss (and potentially*

*rethink) this."*

After reviewing your response, it seems clear that the approach taken for observational data (black lines in Figs. 8 & 9) is resulting in an overestimation of the near-surface cloud frequency of occurrence. You define a minimum threshold of ceilometer backscatter based on Kuma et al. [2021], but it seems you have not attempted to screen for precipitation occurrence (if I have misunderstood this, please clarify!).

Line-specific comments:

**Line 263-264: "The cloud detection algorithm typically identifies observed precipitation as "cloud", whereas the simulated profile does not show any backscattering in the area where precipitation is occurring."**

Does this mean that the precipitation which had erroneously resulted in a cloud detection is then corrected (no backscatter → no cloud)? It seems this is the case for the simulated but not the observed profiles, but again, please clarify.

**Line 264-267: "Upon reviewing the backscatter profiles, certain layers beneath stratocumulus clouds are identified as clouds, potentially consisting of drizzle, snow, fog, or aerosol. Nevertheless, the frequency of such occurrences is insufficient to significantly impact the statistics in a manner comparable to the model bias. Stanford et al. (2023) found ceilometer on Macquarie Island was obscured 2.5 % of the time because of fog."**

Even if fog occurs only 2.5% of the time, you are neglecting to emphasize the extent to which drizzle might be biasing the low-level CFO pictured in Figs. 8 & 9. In Tansey et al. [2022] we estimate small-particle precipitation (drizzle) occurrence to be ~36% at the MICRE site. How is this impacting the observation CFOs in your plots?

My suggestions for minor revisions are as follows:

1) At the very least, discuss this uncertainty in the chosen approach: precipitation, which occurs very often at Macquarie Island, is causing backscatter near the surface to surpass your detection threshold and thus, the observational data's low-level CFO is overestimated (possibly by as much as >30%). It is not sufficient to state that the

frequency of precipitation does not significantly impact the statistics, without providing some quantitative reasoning.

2) Kuma et al. discuss how precipitation may account for disagreement between simulated and observed profiles and they suggest screening for this: "If desired, the attenuated volume backscattering coefficient profiles affected by precipitation can be excluded before the comparison or their fraction determined by visually inspecting the observed attenuated volume backscattering to assess their possible effect on the statistical results."

You might attempt to exclude precipitation, e.g. by defining cloudy bins only above the estimated CBH; neglect bins below CBH that are likely drizzle, and re-calculate the black lines in Figs. 8 & 9.

---

## Author Response (AR2)

**Author Response and Manuscript Revision**

**Zhangcheng Pei**

**September 2023**

Dear Dr. Lebsock,

Please find our response to the second round of Reviewers comments and our revised manuscript.

We have tested the influence of cloud threshold value on cloud detection and found that $6 \times 10^{-6}$ m$^{-1}$ sr$^{-1}$ is more suitable for Macquarie Island than previous default threshold value ($2 \times 10^{-6}$ m$^{-1}$ sr$^{-1}$). This new threshold avoids false detection of boundary layer aerosol but remains satisfactory detection of clouds. We re-ran the ALCF with the new cloud threshold value and found a significant reduction of surface CFO. In addition, we have added additional discussion for the influence of precipitation and fog on observational low-level CFO. We hope that our further explanation and changes satisfy both you and the Reviewers.

Kind regards,

Mr. Zhangcheng Pei

**Response to Reviewer 1:**

**Overall comments:**

**After reviewing your response, it seems clear that the approach taken for observational data (black lines in Figs. 8 & 9) is resulting in an overestimation of the near-surface cloud frequency of occurrence. You define a minimum threshold of ceilometer backscatter based on Kuma et al. [2021], but it seems you have not attempted to screen for precipitation occurrence (if I have misunderstood this, please clarify!).**

Yes, the default minimum threshold value ($2 \times 10^{-6}$ m$^{-1}$ sr$^{-1}$) we used to detect cloud overestimated near-surface CFO, which was influenced by precipitation, fog, and aerosol. We have picked several days when there are a lot of misidentified clouds to test different threshold values. We found that $6 \times 10^{-6}$ m$^{-1}$ sr$^{-1}$ is suitable for Macquarie Island to remove the effect of boundary layer aerosol. The surface CFO was reduced from ~ 60% to ~35%. Our conclusions of CFO regarding radiation biases haven't changed. However, we have not screened for precipitation occurrence as we don't have disdrometer or rain gauge data for this specific period (Sep 2017 – Feb 2018).

**Line-specific comments:**

**Line 263-264: "The cloud detection algorithm typically identifies observed precipitation as "cloud", whereas the simulated profile does not show any backscattering in the area where precipitation is occurring."**

**Does this mean that the precipitation which had erroneously resulted in a cloud detection is then corrected (no backscatter → no cloud)? It seems this is the case for the simulated but not the observed profiles, but again, please clarify.**

The precipitation was not distinguished from clouds in observation as it can produce as strong backscatter as cloud. The model doesn't suffer this problem as it simulates backscatter based on cloud properties such as cloud fraction and liquid/ice mass mixing ratio, but not on precipitation properties. For this reason the simulated profile does not show backscattering for precipitation.

**Line 264-267: "Upon reviewing the backscatter profiles, certain layers beneath stratocumulus clouds are identified as clouds, potentially consisting of drizzle, snow, fog, or aerosol. Nevertheless, the frequency of such occurrences is insufficient to significantly impact the statistics in a manner comparable to the**

**model bias. Stanford et al. (2023) found ceilometer on Macquarie Island was obscured 2.5 % of the time because of fog."**

**Even if fog occurs only 2.5% of the time, you are neglecting to emphasize the extent to which drizzle might be biasing the low-level CFO pictured in Figs. 8 & 9. In Tansey et al. [2022] we estimate small-particle precipitation (drizzle) occurrence to be ~36% at the MICRE site. How is this impacting the observation CFOs in your plots?**

Yes, the small-particle precipitation should be emphasized to be able to increase the low-level CFO. Tansey et al. (2022) use the dataset from April 2016 – March 2017, which is different from our study period for CFO here (September 2017 – February 2018), but the results still can be referred to to show the frequency of precipitation occurrence at Macquarie Island.

**Suggestions:**

**1) At the very least, discuss this uncertainty in the chosen approach: precipitation, which occurs very often at Macquarie Island, is causing backscatter near the surface to surpass your detection threshold and thus, the observational data's low-level CFO is overestimated (possibly by as much as >30%). It is not sufficient to state that the frequency of precipitation does not significantly impact the statistics, without providing some quantitative reasoning.**

We have added the discussion as below.

Line 497: Nevertheless, it is crucial to note that limitations exist in ALCF for reproducing CFO. As mentioned in Section 2.6, ALCF doesn't identify precipitation, which could be classified as cloud in the ceilometer while ignored in the model (Kuma et al.,2021). This may cause an overestimation of CFO near the surface in the ceilometer and potentially amplify the underestimation of low-level CFO in the model. Upon visually inspecting the time series of ceilometer backscatter profiles, certain layers beneath stratocumulus clouds at around 500 m are identified as clouds, potentially consisting of drizzle, snow, or fog. Tansey et al. (2022) has reported an occurrence of 34% and 19% of drizzle in 2016-17 spring and summer at Macquarie Island. Moreover, Stanford et al. (2023) found that ceilometer observations on Macquarie Island were obscured 18 % of the time because of fog, which is also likely to influence the CFO near the surface. Hence, low-level CFO below 500 m should be interpreted cautiously as it could be influenced by the combination of precipitation and fog. Further research that combines lidar/ceilometer with precipitation measurements will be beneficial to the model evaluation. Moreover, more sophisticated algorithms to classify precipitation, fog, and aerosol are suggested to be developed within ALCF.

**2) Kuma et al. discuss how precipitation may account for disagreement between simulated and observed profiles and they suggest screening for this: "If desired, the attenuated volume backscattering coefficient profiles affected by precipitation can be excluded before the comparison or their fraction determined by visually inspecting the observed attenuated volume backscattering to assess their possible effect on the statistical results." You might attempt to exclude precipitation, e.g. by defining cloudy bins only above the estimated CBH; neglect bins below CBH that are likely drizzle, and re-calculate the black lines in Figs. 8 & 9.**

We have tried to exclude precipitation by neglect bins below CBH observed by Vaisala CL51 ceilometer. While we found that, in many situations, the Vaisala CBH is in the middle of the cloud instead of the bottom (Figure 1), which may be due to its algorithm or calibration method. Thus, this method is not so valid for removing the effect of precipitation.

[Figure]

Figure 1: Attenuated volume backscattering coefficient profile with CBH from Vaisala CL51 ceilometer.

As we don't have disdrometer or rain gauge dataset from this period, we are not able to exclude precipitation and fog's effect in this work. We have raised caution about the low-level CFO, and that we should involve precipitation measurements for comprehensive model evaluation for in future studies. Moreover, ALCF is suggested to develop algorithm that can classify precipitation and fog.

**Response to Reviewer 2:**

**In response to concerns raised by Reviewer 1 regarding high low-level COF in Figures 8 & 9, you state the following: "Stanford et al. (2023) found ceilometer on Macquarie Island was obscured 2.5 % of the time because of fog." While Stanford et al. did indeed find the ceilometer was obscured 2.5 % of the time, likely due to fog, they also did a more formal analysis of fog and found 18 % of all profiles to be representative of fog (i.e., high surface RH and completely attenuated layers very close to the surface). I think this should be mentioned since fog is likely influencing your COF at low levels. However, I also think this is okay since fog is still cloud, and you explicitly state your attenuated backscatter coefficient threshold and support it through citation of Kuma et al. (2021). Should you wish, you could show the sensitivity to this threshold (perhaps in an Appendix) showing a plot where the x-axis is the attenuated backscatter coefficient threshold, the y-axis is height, and contours would represent the COF for a given threshold. This is not necessary since you state and support your threshold, but could be an extra supportive measure.**

We have added the discussion of influence of precipitation and fog on low-level COF.

Line 497: Nevertheless, it is crucial to note that limitations exist in ALCF for reproducing CFO. As mentioned in Section 2.6, ALCF doesn't identify precipitation, which could be classified as cloud in the ceilometer while ignored in the model (Kuma et al.,2021). This may cause an overestimation of CFO near the surface in the ceilometer and potentially amplify the underestimation of low-level CFO in the model. Upon visually inspecting the time series of ceilometer backscatter profiles, certain layers beneath stratocumulus clouds at around 500 m are identified as clouds, potentially consisting of drizzle, snow, or fog. Tansey et al. (2022) has reported an occurrence of 34% and 19% of drizzle in 2016-17 spring and summer at Macquarie Island. Moreover, Stanford et al. (2023) found that ceilometer observations on Macquarie Island were obscured 18 % of the time because of fog, which is also likely to influence the CFO near the surface. Hence, low-level CFO below 500 m should be interpreted cautiously as it could be influenced by the combination of precipitation and fog. Further research that combines lidar/ceilometer with precipitation measurements will be beneficial to the model evaluation. Moreover, more sophisticated algorithms to classify precipitation, fog, and aerosol are suggested to be developed within ALCF.

Additionally, we have picked up several days when there are a lot of misidentified cloud to test different threshold values. We found that $6 \times 10^{-6}$ m$^{-1}$ sr$^{-1}$ is suitable for Macquarie Island to remove the effect of boundary layer aerosol. The surface CFO was reduced from ~ 60% to ~35%. Our conclusions of CFO regarding radiation biases haven't changed.